Microbiology
**Spectrum**

# Comparative analysis of the lung microbiota in patients with lung cancer, chronic obstructive pulmonary disease, and community-acquired pneumonia

Jia Xu,[1,2,3] Yingmiao Zhang,[1,4] Lifeng Shi,[1] Hui Wang,[1] Ming Zeng,[5] Zhongxin Lu[1,2,4]

**ABSTRACT**  Respiratory diseases pose a significant global public health challenge. Extensive research indicates that respiratory conditions are influenced by lung microbiota; however, the relationships between alterations in pulmonary microbiota and various respiratory diseases remain unclear. This study explores the characteristics and distinctions of lung microbial communities in patients with lung cancer (LC), chronic obstructive pulmonary disease (COPD), and community-acquired pneumonia (CAP). The research involved 114 patients and employed culturomics and 16S rRNA gene sequencing to analyze bronchoalveolar lavage fluid samples. Through culturomics, 168 bacterial species were identified, with variations in bacterial profiles observed across the different diseases. Sequencing results indicated that the dominant phyla among the three groups were *Bacillota*, *Bacteroidota*, *Pseudomonadota*, *Actinomycetota*, and *Fusobacteriota*, consistent with the culturomics findings. Notably, the CAP group exhibited higher species richness compared to the LC and COPD groups, with significant differences in beta-diversity among the three groups. Specific bacterial genera, such as *Alloprevotella*, *Abiotrophia*, and *Mycoplasma*, were distinguished as indicative taxa for the LC, COPD, and CAP groups, respectively. Utilizing random forest modeling and receiver operating characteristic curve analysis, several key bacterial genera were identified as capable of differentiating between these diseases. The study highlights distinct differences in lung microbiota among patients with LC, COPD, and CAP, potentially serving as a reference for diagnosis, suggesting that disease-specific microenvironments may influence local microbial communities, thus providing evidence for associations between lung microbiota and various respiratory diseases that warrant further investigation.

**IMPORTANCE** The human lung microbial community plays a crucial role in various respiratory diseases by regulating the lung's immune system and maintaining lung homeostasis. However, there is a paucity of comparative studies examining the characteristics of the pulmonary microbiome in common respiratory diseases, such as lung cancer (LC), chronic obstructive pulmonary disease (COPD), and community-acquired pneumonia (CAP). This study aims to explore the differences in lung microbiomes among these conditions. By employing culturomics and 16S rRNA sequencing technology, we identified significant variations in their lung microbiota. Notably, *Alloprevotella*, *Abiotrophia*, and *Mycoplasma* were identified as indicative taxa for the LC, COPD, and CAP groups, respectively. This research is essential for enriching the database of cultivable lung bacteria and investigating the interactions between specific strains and diseases at the species level, and identifying potential biomarkers and therapeutic targets.

**KEYWORDS**  microbiota, lung cancer, chronic obstructive pulmonary disease, community-acquired pneumonia, culturomics, 16S rRNA gene sequencing

Address correspondence to Zhongxin Lu, luzhongxin@zxhospital.com.

Jia Xu and Yingmiao Zhang contributed equally to this article. Author order was determined based on the nature of their contributions to the study and in order of increasing seniority.

The authors declare no conflict of interest.

See the funding table on p. 16.

With the proposal and implementation of the Human Microbiome Project, there has been a growing interest in the role of microbiota in diseases, leading to extensive research in this field (1). Studies have demonstrated the presence of various microbiomes in the lungs, which are involved in regulating the host's pulmonary immune systems and maintaining lung homeostasis (2–6). Various respiratory conditions, such as lung cancer (LC) (3), chronic obstructive pulmonary disease (COPD) (7), community-acquired pneumonia (CAP) (8), cystic fibrosis (9), tuberculosis (10), and asthma (11), exhibit significant differences in their lung microbiota composition. These changes exacerbate dysbiosis and perpetuate a cycle of inflammation and disrupted microbiota (12).

LC is recognized as one of the most burdensome malignant tumors globally and is often referred to as the "top fatal cancer killer." It has also witnessed the fastest growth in incidence rates in China over the past three decades (13). COPD is a leading cause of rising incidence and mortality rates of chronic diseases globally and is emphasized in the Healthy China 2030 Action Plan as a key disease for prevention and control (14). CAP has been identified as one of the major infectious diseases contributing to hospitalization and mortality worldwide, posing a significant public health challenge (15). These three ailments are prevalent respiratory disorders, each with its unique pathogenesis, and they exhibit complex interrelationships (16, 17). Some researchers have proposed that inflammation-induced tissue damage may contribute to the development of LC (18), while other studies have found an association between a history of COPD and an increased risk of LC (19). However, disparities in the pulmonary microbiome profiles of individuals with LC, COPD, and CAP remain to be explored.

In this study, we utilized culturomics and 16S rRNA sequencing to analyze the characteristics of lung microbiota in bronchoalveolar lavage fluid (BALF) samples from patients with LC, COPD, and CAP. We conducted alpha and beta diversity analysis, Linear Discriminant Analysis Effect Size (LEfSe), Random Forest model, and receiver operating characteristic (ROC) curve analysis. Our findings enrich the cultivable bacterial database for patients with LC, COPD, and CAP, facilitate in-depth investigations into host-microbe interactions, and explore the relevant characteristics of key strains in various pulmonary diseases.

## RESULTS

### Baseline characteristics of participants

A total of 114 eligible participants were enrolled in the study, categorized into three groups based on their clinical diagnoses: 42 cases of LC, 32 cases of COPD, and 40 cases of CAP. BALF samples from these participants were cultured and identified using culturomics. Subsequent 16S rRNA gene sequencing was performed on a final set of 48 qualified samples that met our quality control criteria, which included sufficient residual volume, adequate DNA concentration, and passing library quality inspection. These sets comprised 18 cases of LC, 16 cases of COPD, and 14 cases of CAP. The baseline characteristics of participants are presented in Table 1.

### Characteristics of bacteria isolated from BALF via culturomics

By culturing all 114 BALF samples with four different conditions, a total of 3,346 colonies were identified, consisting of 5 phyla, 13 classes, 25 orders, 39 families, 58 genera, and 168 species. Among these, 113 species were found in the LC group, 109 species in the COPD group, and 102 species in the CAP group. Additionally, 61 species were isolated from all three groups, with 29, 25, and 19 unique species identified in the LC, COPD, and CAP groups, respectively (Fig. 1A). The 168 identified bacterial species were classified into five main phyla, with *Bacillota*, *Pseudomonadota*, and *Actinomycetota* collectively representing the majority (>85%) of the cultivated community (Fig. 1B). For taxonomic analysis of the isolated bacteria, we characterized the bacterial community composition at various taxonomic levels. Figure 1C displays the five phyla, the top 22 dominant families, and the top 27 dominant genera of the pulmonary microbiota in patients

**TABLE 1** Baseline characteristics of participants[a]

| Variable | Culturomics | | | 16S rRNA sequencing | | |
|---|---|---|---|---|---|---|
| | LC | COPD | CAP | LC | COPD | CAP |
| | (*n* = 42) | (*n* = 32) | (*n* = 40) | (*n* = 18) | (*n* = 16) | (*n* = 14) |
| Age, mean (±SD) | 65.1 ± 9.4 | 68.94 ± 9.14 | 50.6 ± 19.5 | 64.4 ± 9.2 | 70.1 ± 9.5 | 48.6 ± 18.7 |
| Male | 28 (66.7%) | 24 (75.0%) | 18 (45%) | 14 (77.8%) | 13 (81.3%) | 7 (50%) |
| Female | 14 (33.3%) | 8 (25.0%) | 22 (55%) | 4 (22.2%) | 3 (18.7%) | 7 (50%) |
| Current or former smoker | 17 (40.5%) | 24 (75.0%) | 9 (22.5%) | 10 (55.6%) | 14 (87.5%) | 4 (28.6%) |
| Never smoker | 25 (59.5%) | 8 (25.0%) | 31 (77.5%) | 8 (44.4%) | 2 (12.5%) | 10 (71.4%) |
| Adenocarcinoma | 27 (64.3%) | NA | NA | 10 (55.6%) | NA | NA |
| Squamous cell carcinoma | 9 (21.4%) | | | 5 (27.8%) | | |
| Small cell lung cancer | 3 (7.15%) | | | 3 (16.6%) | | |
| Unidentified | 3 (7.15%) | | | NA | | |
| Metastatic | 25 (59.5%) | NA | NA | 12 (66.7%) | NA | NA |
| Non-metastatic | 17 (40.5%) | | | 6 (33.3%) | | |

[a]Data are presented as mean (±SD) for continuous variables and total number (percentage) for categorical variables. NA indicates not applicable.

with LC, COPD, and CAP. These results indicate diversity in the composition of the lung microbiota among patients with LC, COPD, and CAP. The number of bacterial species isolated from each sample under different culture conditions is listed in Table S1.

Meanwhile, we evaluated the distribution of bacteria isolated from the three groups of samples at the phylum level under different culture conditions. The total number of bacteria isolated from the three groups of BALF samples through direct inoculation culture (carbon dioxide culture and microaerobic culture) was higher than that from pre-incubation culture (aerobic bottle and anaerobic bottle). Among the three groups, the LC and COPD groups demonstrated the highest diversity of bacterial species isolated under microaerobic culture conditions, with 67 and 53 species, respectively. In contrast, the CAP group yielded the highest number of bacterial species, totaling 58 species, under carbon dioxide culture conditions. The dominant phyla in all three groups were *Bacillota*, followed by *Pseudomonadota* and *Actinomycetota*. Notably, in the LC and COPD groups, species from the phylum *Fusobacteriota* were exclusively isolated under microaerophilic culture conditions, whereas in the CAP group, they could be isolated from both carbon dioxide culture and microaerobic culture environments (Fig. S1A). The number of bacterial species isolated under all four culture conditions in the three groups was 16, 14, and 13, respectively. However, a significant number of bacteria were still isolated only under specific culture conditions (Fig. S1B). This indicates that combining multiple culture conditions can achieve effective separation and cultivation of bacteria. Additionally, an analysis of the isolation frequencies of individual bacteria in the three groups was conducted. Figure S1C illustrates the top 20 bacterial species with the highest isolation frequencies within each group. *Streptococcus mitis* (78.57%) exhibited the highest isolation frequency in the LC group, *Streptococcus parasanguinis* (68.75%) in the COPD group, and *Schaalia odontolyticus* (82.50%) in the CAP group. Furthermore, we isolated five strains of bacteria that, although previously obtained from environmental sources, have now been isolated from clinical patients for the first time (Table S2).

## Lung microbiome profiles analyzed by 16S rRNA gene sequencing

For 16S rRNA gene sequencing, a total of 3,726,398 raw reads were generated from the 48 samples, which included 18 LC samples, 16 COPD samples, and 14 CAP samples. These reads were then processed, resulting in 3,713,184 high-quality reads (1,599,354 for the LC group, 1,218,410 for the COPD group, and 895,420 for the CAP group), with an average of 77,358 reads per sample. Moreover, the analysis of the pulmonary microbiota revealed the presence of 3,780 operational taxonomic units (OTUs), with an average coverage of 99.87%, indicating comprehensive detection of the pulmonary bacterial community. According to the Venn diagram shown in Fig. 2A, there were 986 shared OTUs among the

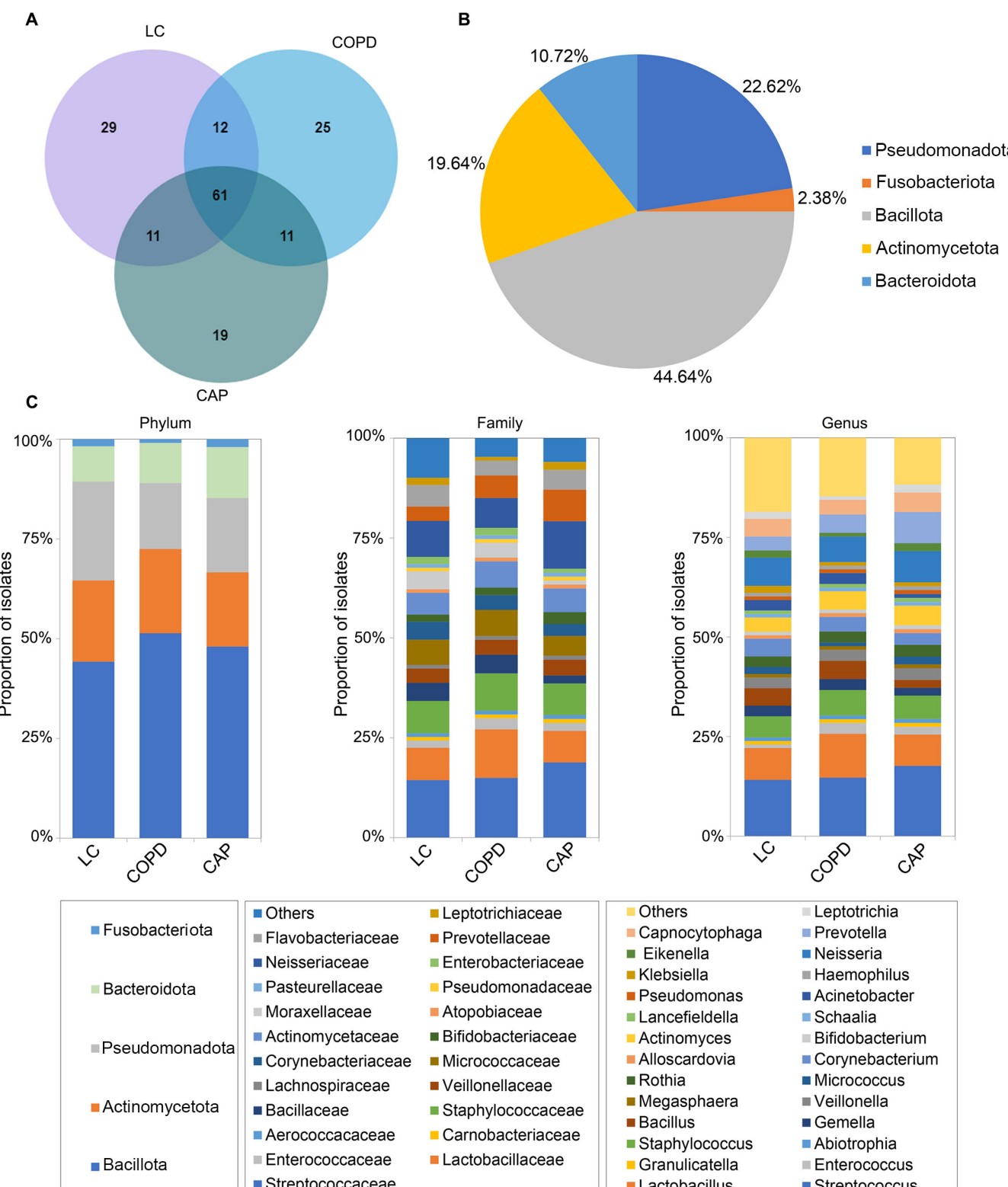

**FIG 1** Bacteria identified from the bronchoalveolar lavage fluid (BALF) samples using culturomics. (A) Venn diagram showing culturable bacterial species among three groups. (B) Proportion of 168 bacterial species isolated from the BALF samples listed according to phylum. (C) Relative proportions of isolated bacterial phyla, families, and genera in lung microbiota of patients with lung cancer (LC), chronic obstructive pulmonary disease (COPD), and community-acquired pneumonia (CAP).

three groups, while 474, 255, and 1,170 OTUs were unique to patients with LC, COPD, and CAP, respectively. Bacterial OTUs were classified into 28 phyla, 63 classes, 155 orders, 272 families, 560 genera, and 676 species.

At the phylum level, the main constituents in the three groups were *Bacillota* (LC, 31.8%; COPD, 35.3%; CAP, 45.4%), *Bacteroidota* (LC, 27.2%; COPD, 23.0%; CAP, 25.8%), and *Pseudomonadota* (LC, 26.3%; COPD, 21.0%; CAP, 14.7%), which together comprised 79.4%–85.8% of the detected phyla. In the COPD and LC groups, *Actinomycetota* (13.0% and 6.6%) was the fourth most abundant phylum, while in the CAP group, *Fusobacteriota* (3.1%) occupied this position (Fig. 2B). Regarding the analysis at the genus level, significant differences were observed in the composition of microbial communities among the three groups. Specifically, the top 10 genera in the LC group were *Streptococcus*, *Prevotella_7*, *Haemophilus*, *Alloprevotella*, *Pseudomonas*, *Veillonella*, *Prevotella*, *Porphyromonas*, *Moraxella*, and *Neisseria*. In the COPD group, the top 10 genera included *Streptococcus*, *Porphyromonas*, *Prevotella_7*, *Veillonella*, *Domibacillus*, *Pseudomonas*, *Corynebacterium*, *Rothia*, *Haemophilus*, and *Escherichia-Shigella*. The CAP group showcased *Mycoplasma*, *Prevotella_7*, *Streptococcus*, *Veillonella*, *Prevotella*, *Granulicatella*, *Alloprevotella*, *Actinobacillus*, *Porphyromonas*, and *Fusobacterium* as the top 10 genera. It is worth noting that *Streptococcus*, *Prevotella_7*, *Veillonella*, and *Porphyromonas* were present in the top 10 of each group (Fig. 2C).

## Biodiversity in the lung microbiota in patients with LC, COPD, and CAP

The Shannon and Chao1 indices were utilized to evaluate the abundance and diversity of microbes in the three groups. There were no significant differences in the Shannon index ($P = 0.28$; Fig. 3A) among the three groups. However, Fig. 3B illustrates that the CAP group had significantly higher values for the Chao1 index, compared to the other two groups. The results from the rarefied data fully corroborate our original findings (Fig. S2). Bacterial beta diversity analysis was conducted by generating principal coordinate analysis (PCoA) plots using Bray–Curtis and Weighted Unifrac distances, which were employed to evaluate the overall differences in the pulmonary microbiota across the three groups. The results demonstrated significant distinctions among the three groups (Bray–Curtis: $R = 0.2862$, $P = 0.001$; Weighted Unifrac: $R = 0.3222$, $P = 0.012$; Fig. 3C and D). Taken together, our findings indicate variations in bacterial alpha and beta diversity in the pulmonary microbiota among the LC, COPD, and CAP groups.

## Combined analysis of distinct taxa and key pathogenic bacteria in BALF samples from LC, COPD, and CAP groups

To investigate the microbial composition in BALF among lung disease groups, the LEfSe and a Random Forest model were employed. First, the LEfSe was used to identify taxa with varying abundances across the three groups. The resulting Linear Discriminant Analysis (LDA) score histograms (Fig. 4A) and cladograms (Fig. 4B) revealed 29 distinct taxa at different taxonomic levels. Notably, at the genus level, *Alloprevotella*, *Abiotrophia*, and *Mycoplasma* were identified as indicative taxa for the LC, COPD, and CAP groups, respectively. Furthermore, we performed a species-level analysis of the lung microbiota, which provided valuable insights into the significant microbiota that differentiate among the three groups (Fig. S3A and B). In addition to identifying specific taxa with LEfSe, we conducted a Random Forest analysis to screen for key pathogenic bacteria distinguishing the three sample groups. This model revealed crucial insights into the bacterial composition, with *Treponema*, *Mycoplasma*, *Tannerella*, *Aquabacterium*, and *Rothia* exhibiting high accuracy scores (Fig. 4C). To evaluate the predictive capability of the Random Forest model, we plotted ROC curves and calculated the area under the curve (AUC) values. Our findings indicated that *Treponema* (AUC = 0.877, $P = 0.002$), *Oribacterium* (AUC = 0.797, $P = 0.0032$), and *Alloprevotella* (AUC = 0.767, $P = 0.0079$) demonstrated excellent specificity and sensitivity in differentiating between the LC and COPD groups (Fig. 4D). For the differentiation between the LC and CAP groups, *Tannerella* (AUC = 0.899, $P = 0.0001$) was identified as the best predictive genus, followed by *Aerococcus* (AUC = 0.814, $P = 0.0027$)

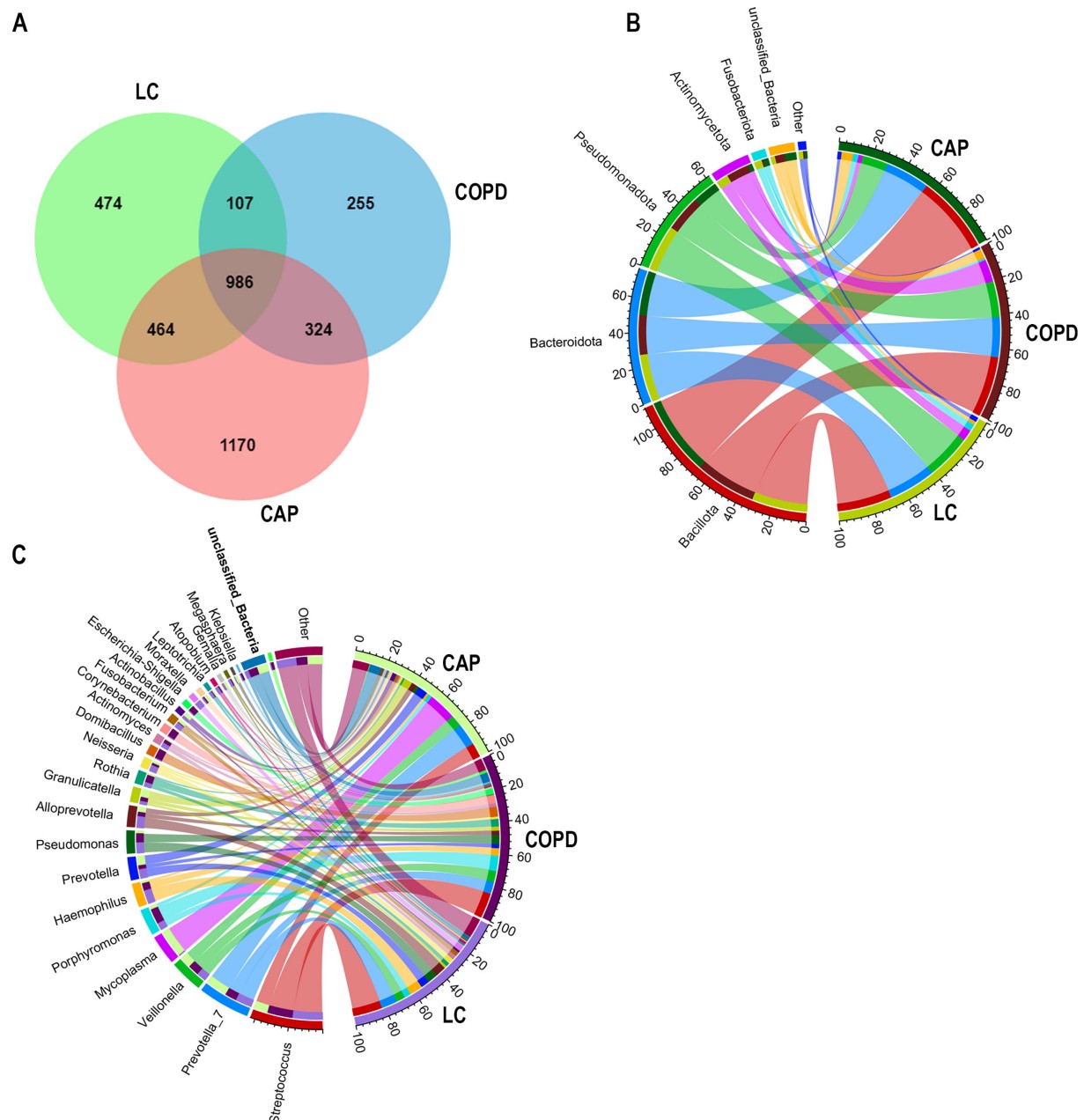

**FIG 2** Structural composition of the microbiota in the lung cancer (LC), chronic obstructive pulmonary disease (COPD), and community-acquired pneumonia (CAP) groups. (A) Venn diagram illustrating the operational taxonomic units in lung microbiota among three groups. (B) Circos showing the composition of lung microbiota at the phylum level in the three groups. (C) Circos showing the composition of lung microbiota at the genera levels in the three groups.

and *Treponema* (AUC = 0.794, *P* = 0.0049; Fig. 4E). In distinguishing COPD from CAP, *Citrobacter* emerged as the superior genus (AUC = 0.810, *P* = 0.0039), with *Enterococcus* (AUC = 0.799, *P* = 0.0053) and *Cutibacterium* (AUC = 0.788, *P* = 0.0073) closely following (Fig. 4F).

## An in-depth analysis of the lung microbiome in LC patients with varied clinicopathological profiles

For patients with LC, we further categorized them into adenocarcinoma (ADC) and squamous cell carcinoma (SCC) groups, as well as metastatic and non-metastatic groups, based on their pathological diagnoses and distant metastasis status. We conducted 16S

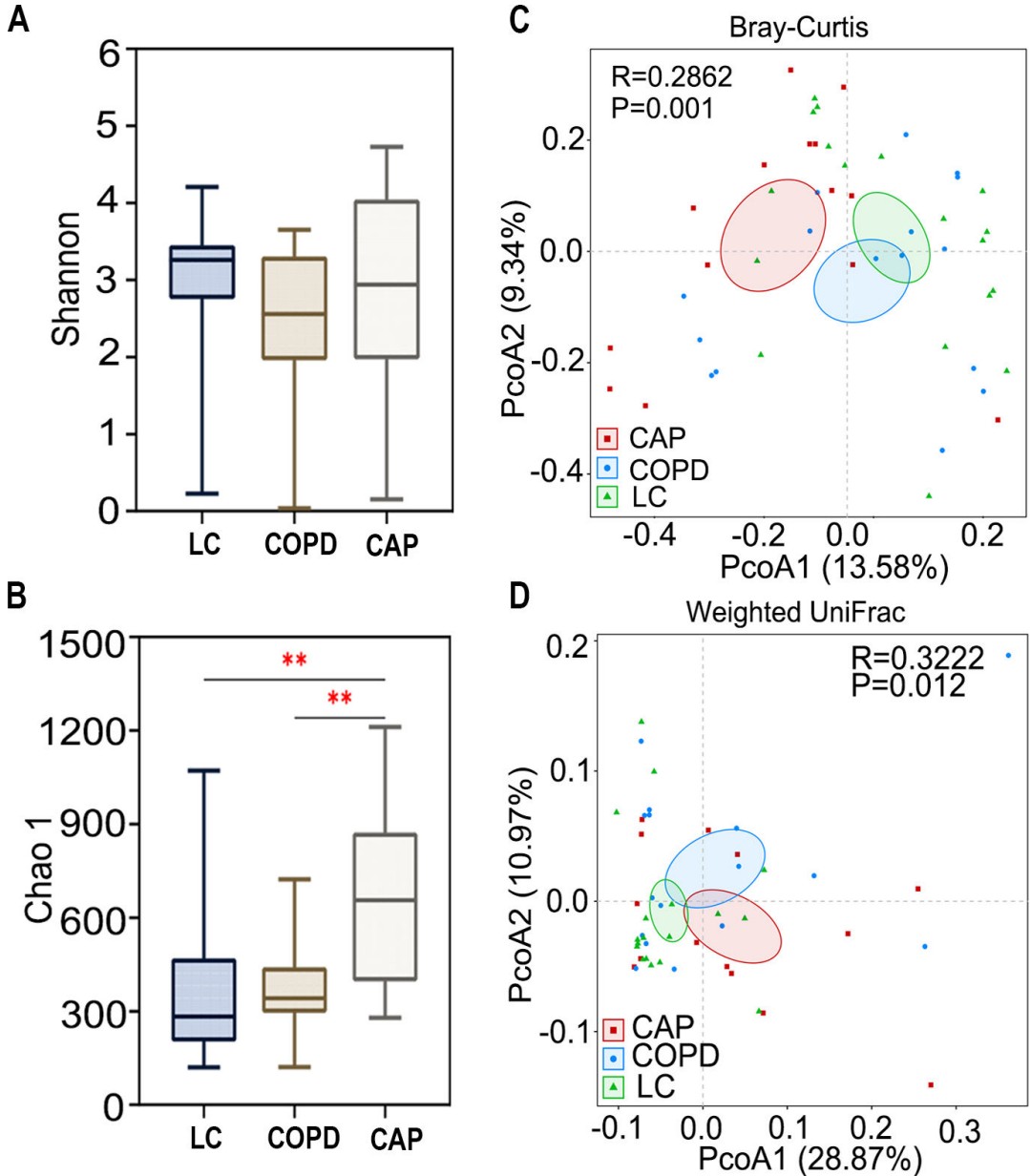

**FIG 3** Biodiversity in the lung microbiota in patients with lung cancer (LC), chronic obstructive pulmonary disease (COPD), and community-acquired pneumonia (CAP). (A–D) The alpha diversity was assessed using the Shannon index (A) and the Chao1 index (B). ***, *P* < 0.001; **, *P* < 0.01. (C and D) Beta diversity was analyzed using Bray–Curtis (C) and Unweighted UniFrac distance (D) (Bray–Curtis: *R* = 0.2862, *P* = 0.001; Weighted Unifrac: *R* = 0.3222, *P* = 0.012). The red, blue, and green dots represent CAP, COPD, and LC samples, respectively.

rRNA sequencing analysis to investigate the lung microbiome. By comparing the relative abundance of bacteria at the genus level between the ADC and SCC groups, we observed that *Streptococcus* (21.7%) was the predominant genus in the ADC group, followed by *Prevotella_7* (12.4%) and *Alloprevotella* (6.6%). In contrast, the SCC group exhibited the highest relative abundance of *Pseudomonas* (24.8%), while *Streptococcus* (12.9%) and *Alloprevotella* (7.3%) ranked second and third, respectively (Fig. 5A). For bacterial alpha diversity analysis, the Shannon index (*P* = 0.724) and Chao1 index (*P* = 0.781) showed similar diversity between the two groups (Fig. S4A and B). Beta diversity, assessed using Bray-Curtis distances (*R* = 0.2885, *P* = 0.269), also revealed no significant differences between the two groups (Fig. S4C). Subsequently, we performed a LEfSe analysis to identify microbial biomarkers that distinguish between the two groups. The

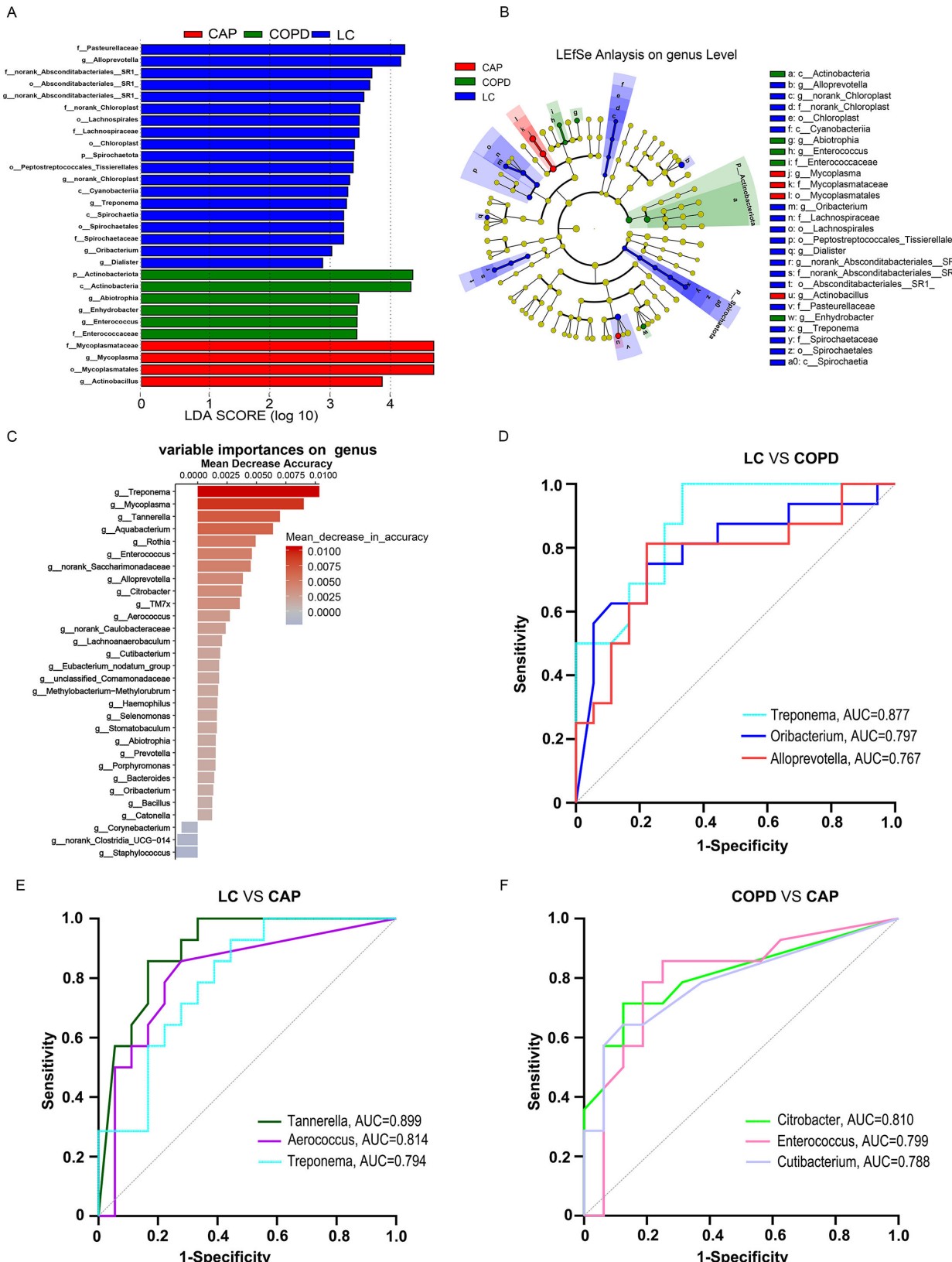

FIG 4 Combined analysis of distinct taxa and key pathogenic bacteria in bronchoalveolar lavage fluid samples from lung cancer (LC), chronic obstructive pulmonary disease (COPD), and community-acquired pneumonia (CAP) groups. (A) The linear discriminant analysis (LDA) scores of taxa presented the differences in microbiome composition among the three groups. (B) The taxonomic cladogram generated by linear discriminant analysis effect size (LEfSe) analysis on the (Continued on next page)

**Fig 4 (Continued)**

genus level showed the changes of microbiome in patients with LC, COPD, and CAP. (C) The genera that contributed the most to the model are ranked by mean decrease accuracy. (D) Receiver operating characteristic (ROC) analysis indicated the potential diagnostic value in differentiating LC and COPD. (E) ROC analysis indicated the potential diagnostic value in differentiating LC and CAP. (F) ROC analysis indicated the potential diagnostic value in differentiating COPD and CAP.

results indicated that *Prevotella_7 melaninogenica*, *Haemophilus parainfluenzae*, and *Haemophilus influenzae* may serve as potential biomarkers in both the ADC and SCC groups (Fig. 5B). The AUC was utilized to measure the predictive accuracy of the three species identified by LEfSe. Our findings showed that *Prevotella_7 melaninogenica* (AUC = 0.860, $P$ = 0.0275) and *Haemophilus parainfluenzae* (AUC = 0.840, $P$ = 0.0373) had higher AUC values (Fig. 5C).

Previous studies have suggested a relationship between the lung microbiome and the development of distant metastasis in LC (20, 21). Therefore, the differences in microbial composition between metastatic and non-metastatic groups were assessed using the same methodology. Analysis of the abundance at the genus level revealed that the top two genera in both groups were *Streptococcus* (metastatic: 19.5% and non-metastatic: 13.0%) and *Prevotella_7* (metastatic: 9.4% and non-metastatic: 10.9%). The third most abundant genus in the metastatic group was *Haemophilus* (8.8%), whereas in the non-metastatic group, it was *Alloprevotella* (8.1%; Fig. 5D). The results of the diversity analysis employing the Shannon index, Chao1 index, and Bray-Curtis distances suggest that there is no statistically significant variance in alpha and beta diversity between the two groups ($P$ = 0.130 for the Shannon index; $P$ = 0.349 for the Chao1 index; R = 0.2904, $P$ = 0.066 for Bray-Curtis distances; Fig. S4D through F). It was concluded that there was no link between the diversity of the lung microbiome and the presence of distant metastasis in individuals diagnosed with LC (22). A LEfSe analysis was conducted to thoroughly assess the dissimilarities between the metastatic and non-metastatic groups. We observed significant variations in abundance between the 2 groups for 11 distinct taxa at different levels. Of these, four taxa showed differential abundance at the species level, underscoring their importance in distinguishing between the two groups. A higher abundance of *Anaeroglobus geminatus* and *Dialister invisus* was detected in the metastatic group, while *Prevotella nanceiensis* and *Capnocytophaga gingivalis* were found to be more abundant in the non-metastatic group (Fig. 5E). According to ROC analysis, *Dialister invisus* (AUC = 0.833, $P$ = 0.0246) exhibited moderate diagnostic value in differentiating between LC patients with metastasis and those without, whereas *Capnocytophaga gingivalis* (AUC = 0.736, $P$ = 0.1113) did not demonstrate statistical significance (Fig. 5F).

## DISCUSSION

Previous investigations have established a strong association between alterations in lung microbial communities and the progression of diverse pulmonary diseases (5, 23–25). This highlights their critical role in the pathogenesis and development of these diseases through their impact on host inflammation, immune response, and metabolic pathways (3, 26, 27). In this study, we explored and analyzed the lung microbiomes of patients with LC, COPD, and CAP using culturomics and 16S rRNA sequencing, revealing significant differences in their lung microbiota.

By employing culturomics, bacteria can be identified at the species level, significantly enriching the human cultivable bacterial database (28, 29). Reports indicate that the number of bacterial species isolated and cultured from humans increased from 2,776 to 3,253 over 2 years from 2018 to 2020, with the majority of these species (63%) identified through culturomics (30). Some studies investigated the lung microbiota and found that it predominantly consists of *Bacillota*, *Actinomycetota*, *Pseudomonadota*, *Bacteroidota*, and *Fusobacteriota* at the phylum level (31–33), which aligns with our findings. One advantage of culturomics is its ability to isolate viable bacteria, providing the conditions for subsequent research on bacterial strains (28). We have identified five bacterial strains that were isolated from humans for the first time through culturomics, and further

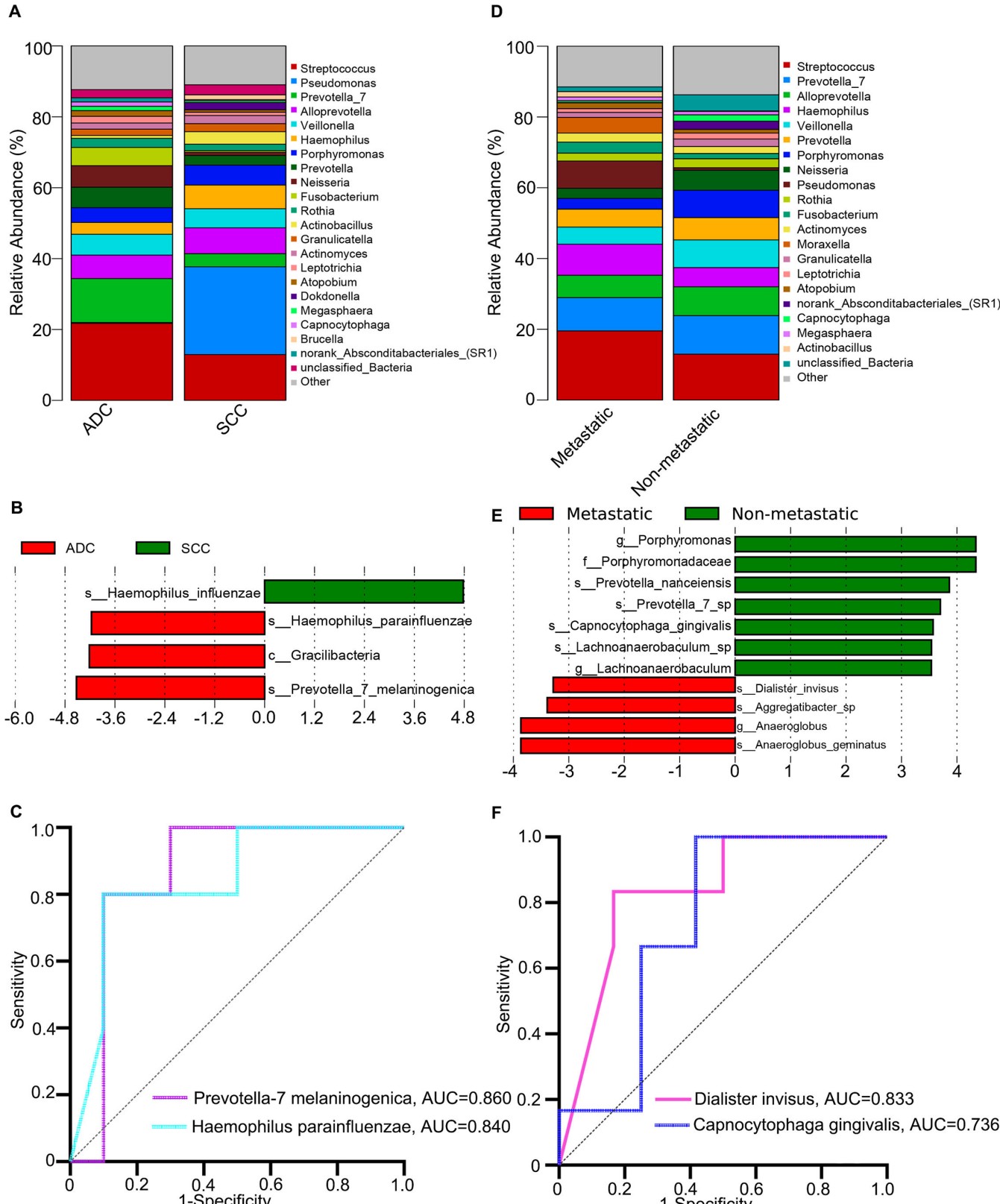

FIG 5 Characterization of lung microbiota in different subtypes of lung cancer (LC). (A) Taxonomic composition at the genus level in the adenocarcinoma (ADC) and squamous cell carcinoma (SCC) groups. (B) Linear Discriminant Analysis Effect Size (LEfSe) was employed to identify bacterial microbiota that significantly differed between ADC and SCC. (C) Receiver operating characteristic (ROC) analysis indicated the potential diagnostic value for differentiating between ADC (Continued on next page)

Fig 5 (Continued)

and SCC. (D) Taxonomic composition at the genus level in the metastatic and non-metastatic groups. (E) LEfSe was utilized to identify bacterial microbiota that significantly differed between metastatic and non-metastatic LC. (F) ROC analysis indicated the potential diagnostic value for differentiating between metastatic and non-metastatic LC.

investigation into their virulence and pathogenicity will yield significant insights. It is well known that certain bacteria can only be cultivated under microaerobic conditions; nonetheless, few researchers in the field of culturomics have considered microaerobic conditions (31, 34). By incorporating microaerobic cultivation conditions, our research revealed a significant increase in the number of cultivable bacteria in the LC, COPD, and CAP groups, by 14.1%, 12.4%, and 9.7%, respectively. This underscores the essentiality of utilizing microaerobic conditions to augment the variety of cultivable bacterial species, serving as a valuable reference for future research in this field.

A notable finding from our culturomics approach was the high isolation frequency of commensal oral bacteria, such as *Streptococcus mitis* and *Schaalia odontolyticus*, across all patient groups. This observation aligns with the concept of the "adapted island model" of lung biogeography, where the lung microbiome is continuously seeded by micro-aspiration from the upper respiratory tract but is subsequently shaped by local host conditions (35, 36). The pervasive presence of these taxa could therefore be interpreted in several ways: as potential procedural contamination, as evidence of normal aerodiges-tive tract communication, or as a sign of pathological dysbiosis. While we employed rigorous bronchoscopic sampling and oral hygiene protocols to minimize contamination, we cannot definitively exclude its contribution. We believe that in the diseased lung, impaired mucosal immunity and mucociliary clearance disrupt the equilibrium, allowing for the enrichment and persistence of orally derived bacteria. This "loss of specificity" and expansion of commensals may thus be a feature of the compromised lung microenviron-ment, potentially contributing to chronic inflammation and disease progression rather than merely being a passive reflection of anatomy (37–39).

We observed significant differences in the Chao1 index among the three groups. In contrast, there were no differences in the Shannon index. Notably, the CAP group exhibited not only a higher microbiota richness but also the greatest heterogeneity in microbial composition. This suggests that the CAP lung microenvironment may support a wider array of microbial communities, potentially due to the diverse etiologies and highly variable host immune responses that characterize this syndrome (40–42). This inherent etiological heterogeneity likely explains the dispersed pattern we observed, suggesting that there is no single "CAP microbiome" but rather a spectrum of microbial states associated with the syndrome. This insight underscores the complexity of CAP and highlights the potential of the lung microbiome to reveal sub-phenotypes within this patient population, which could have implications for personalized treatment strategies in the future. However, opinions on this issue vary. Some researchers suggest that the diversity of lung microbiota in LC patients is essentially similar to that in individuals with benign lung diseases (22, 43, 44), while others argue that the lung microbiota diversity in LC patients is higher than that in those with benign lung diseases (45–47). The findings of these studies are inconsistent, and we posit that multiple factors, including geographical location, sample type and quantity, disease classification, and underlying health conditions, may have contributed to these discrepancies.

Findings from a follow-up study involving 227 patients with liver cirrhosis indicate that *Alloprevotella* has the potential to predict the progression of liver cell carcinoma in high-risk individuals with liver cirrhosis (48). Jiao et al. observed notable enrichment of *Alloprevotella* in patients diagnosed with thyroid cancer (49). Our current study indicates that there is a certain association between the genus *Alloprevotella* and LC. However, the exact mechanism of action of *Alloprevotella* in the development of LC has not been thoroughly investigated. Numerous studies have examined the role of lung microbiota in the onset of LC, suggesting that bacteria may stimulate chronic inflammation by enhancing pro-inflammatory mediators, leading to the proliferation of airway epithelial

cells, ultimately resulting in cellular transformation and tumor development (3, 50–52). Through a series of experiments, Jin et al. discovered that dysregulation of the lung microbiota can induce proliferation and activation of γδT cells, resulting in the secretion of IL-17 and other effector molecules, thereby facilitating inflammation and tumor cell proliferation (53). Subsequent studies should further explore the role of lung microbiota in LC to improve diagnostic and therapeutic strategies for patients.

Results from a study conducted in Taiwan indicated a significant correlation between the severity of COPD and genera such as *Prevotella*, *Rothia*, *Neisseria*, *Porphyromonas*, and *Veillonella* (54). However, our study found that *Abiotrophia* exhibited a significantly higher relative abundance in the COPD group, suggesting its potential as a candidate biomarker for COPD. It is worth noting that our study utilized BALF samples, whereas the previously mentioned studies focused on sputum samples. This difference may influence the interpretation of results due to factors such as patient oral hygiene and oral diseases. Meanwhile, an investigation into the microbial community characteristics associated with oral cancer demonstrated a marked reduction in the abundance of *Abiotrophia* in the cancer group compared to the healthy control group (55). Researchers performed a secretome analysis and assessed the virulence of *Abiotrophia defectiva* within its genus, identifying a substantial number of putative virulence factors and over 20 potential virulence-associated proteins (56). Some studies have also shown that nutritionally variant streptococci, including *Abiotrophia defectiva*, are an important cause of bacteremia and infective endocarditis associated with significant morbidity and mortality (57). Although the role of *Abiotrophia* in COPD is less defined, its enrichment in our COPD cohort suggests that it may contribute to the chronic inflammatory and infectious exacerbations that characterize this disease, potentially through similar pro-inflammatory mechanisms as seen in systemic infections. This makes it a compelling candidate for future research into COPD pathogenesis. Further research is necessary to elucidate the reasons for the prominence of *Abiotrophia* in COPD and its potential effects on the condition.

*Mycoplasma pneumoniae* induces host cells to produce interleukin-8, tumor necrosis factor-α, and other pro-inflammatory cytokines (50). Studies have shown that among children aged five and above who were hospitalized due to community-acquired pneumonia, *Mycoplasma pneumoniae* was the most commonly detected bacterium (51). Some studies have also found that *Mycoplasma pneumoniae* is a key pathogen in adult CAP (57). This is highly consistent with our observation of *Mycoplasma* as an indicative taxon in the CAP group. Its ability to drive a robust inflammatory response provides a plausible mechanistic explanation for the acute clinical presentation of pneumonia. Furthermore, its common detection in pediatric CAP populations underscores its clinical relevance as a key respiratory pathogen, as confirmed by our data in an adult cohort. However, divergent opinions have been expressed by many scholars. A study from Sweden proposed that *Streptococcus pneumoniae* is the leading cause of CAP, with viruses and *Haemophilus influenzae* ranking closely behind (58). Additionally, Zhan et al. investigated the lung microbiota of CAP patients through metagenomic next-generation sequencing, revealing that *Corynebacterium*, *Mycobacterium*, *Streptococcus*, *Klebsiella*, and *Acinetobacter* are prevalent pathogens in CAP (59). As is well known, risk factors for CAP include alcoholism, crowded living conditions, age over 70, immunosuppression, hospitalization, and close contact with children (60, 61), all of which may influence the lung microbiota composition in CAP patients to some extent.

This study has several important limitations. First, as a single-center exploratory study with a limited sample size, the generalizability of our findings may be constrained by geographic, lifestyle, and comorbid factors, which could explain discrepancies with prior reports. Consequently, the proposed associations of *Alloprevotella*, *Abiotrophia*, and *Mycoplasma* with specific diseases remain preliminary and associative; they require validation in larger, multi-center cohorts and mechanistic investigation at cellular or animal levels to establish causal roles in pathogenesis. Second, a fundamental interpretive challenge inherent to BALF-based studies must be acknowledged. While we

observed distinct microbial profiles among disease groups, we cannot definitively disentangle the extent to which these signals arise from genuine differences in the lung microbiota versus gradients of oropharyngeal contamination during sampling. Future studies incorporating rigorous controls (e.g., simultaneous oral wash samples and serial dilutions) are essential to resolve this critical issue and confirm the lung-specific origin of these associations.

## Conclusion

In summary, this study conducted a comparative analysis of the overall structure and composition of lung microbiota in patients with LC, COPD, and CAP using culturomics and 16S rRNA sequencing. The results revealed distinct differences in bacterial communities among the three groups. Our study is of great significance for enhancing the database of cultivable lung bacteria and investigating the interactions between specific strains and diseases at the species level. Furthermore, this research provides evidence for associations between lung microbiota and various respiratory diseases that warrant further investigation.

## MATERIALS AND METHODS

### Patient recruitment

This study enrolled 114 patients who were admitted to the Central Hospital of Wuhan and underwent bronchoscopy between June 2023 and January 2024. The cohort included 42 patients with newly diagnosed LC, 32 patients with COPD, and 40 patients with CAP. All diagnoses were made based on clinical manifestations, chest imaging, pulmonary function tests, cytology or histology, and serological laboratory tests in accordance with established clinical diagnostic guidelines. Patients were excluded if they met any of the following criteria: (i) aged below 18 years; (ii) pregnant; (iii) received probiotics, antibiotics, or corticosteroid treatment within the past month; (iv) underwent cancer surgery, radiation, or chemotherapy, or long-term immunosuppressive therapy; (v) had other active or known infections. The baseline clinical characteristics of the participants were recorded, including age, sex, smoking status, and tumor types.

### Sample collection

BALF was collected by experienced physicians using the optical fiber bronchoscopy procedure, which minimizes contamination from the upper respiratory tract and oral microbial flora. Before the procedure, each patient signed an informed consent form and received a detailed explanation of the methods, objectives, and potential risks involved. Specifically, detailed oral hygiene instructions were provided to patients before the collection to reduce the influence of oral microorganisms. Additionally, subjects were administered lidocaine as a topical anesthetic via a nebulizer before bronchoscopy, followed by sedation with midazolam and fentanyl. The fiberoptic bronchoscope was then inserted into the targeted lung segment, with the tip positioned in the opening of the bronchial branch. A slow infusion of 0.9% sterile physiological saline was administered through the bronchial biopsy channel, and aspiration was performed to collect the BALF. The freshly collected BALF was promptly transferred into aseptic containers and transported to the laboratory within 30 min. After culturomics was completed, the samples were stored at −80°C for subsequent DNA extraction.

### Culturomics

Direct inoculation and pre-incubation techniques were employed to culture the BALF samples. Single colonies were isolated and purified through subculturing. Subsequently, colony identification was conducted using matrix-assisted laser desorption/ionization time-of-flight mass spectrometry (MALDI-TOF MS) and 16S rRNA sequencing to identify

all cultivable bacterial species present in each sample. In each experiment, negative control samples were established to detect potential laboratory contamination. These control samples were devoid of any target biological materials, thereby ensuring that environmental contamination would not affect the results. Throughout the sample processing and cultivation phases, a sterile laboratory environment was maintained to minimize the introduction of external microbes.

### Direct inoculation

The BALF samples were serially diluted (10-fold dilution) from $10^{-1}$ to $10^{-10}$. From each of the three selected dilutions ($10^{-1}$, $10^{-5}$, and $10^{-10}$), 100 μL of the sample was evenly spread onto Columbia blood agar plates (Guangzhou Dijing Microbial Technology Co., Ltd., Guangzhou, China). Each dilution was inoculated on two plates, which were then incubated separately under two different conditions: one set at 35°C with 5% $CO_2$ for carbon dioxide cultivation and the other at 35°C with a microaerobic atmosphere of 5% $O_2$, 10% $CO_2$, and 85% $N_2$. The incubation lasted for 18–24 h to maximize the chances of culturing a diverse range of taxa.

### Pre-incubation

A total of 0.5 mL of BALF was added to both aerobic and anaerobic blood culture bottles supplemented with 5% sterile, defibrinated sheep blood. After incubating in a fully automated blood culture instrument (VersaTREK, Thermo Fisher Scientific Inc., MA, USA) for 3 days, the liquid in the blood culture bottles was diluted and spread onto Columbia blood agar plates using the method previously described. Subsequently, the plates were incubated under aerobic conditions for 24 h or anaerobic conditions for 72 h.

### Isolation and purification of the colonies

Based on the color, size, shape, transparency, wetness, and edge uniformity of single colonies, all morphologically distinct colonies were selected from the plates and sub-cultured under their respective conditions to achieve purification of the isolates. It was essential to select as many distinct single colonies as possible, and there was no limit to the number of colonies selected from each plate.

### Identification of the colonies

Bacterial identification was performed using MALDI-TOF MS (Bruker Daltonik GmbH, Germany). A spectral score of ≥1.80 indicated a reliable profiling result; otherwise, the identification was considered unsuccessful, and further analysis was conducted via 16S rRNA sequencing with universal primers (27F, 5′-AGTTTGATCCTGGCTCAG-3′; 1492R, 5′-GT ATTGCCGCGGCTGCTG-3′). The amplification was performed on the C1000 Thermal Cycler (Bio-Rad Laboratories, Hercules, CA, USA), and the product was purified and subjected to bidirectional Sanger sequencing on the Applied Biosystems 3730XL platform (Thermo Fisher Scientific Inc., MA, USA). The similarity between the 16S rRNA gene sequences of the isolates and those of other organisms was compared to all sequence data in GenBank using the NCBI BLAST algorithm (62). In the case where the 16S rRNA gene sequence exhibited less than 98.65% similarity to known species, it was deemed to be a potentially new species (63). Additionally, MEGA XI software was employed to construct a phylogenetic tree to clarify the taxonomic status of the isolated bacteria (64).

### DNA extraction and 16S rRNA gene amplification

After bringing the previously frozen BALF samples to room temperature, centrifugation was carried out at 10,000 × $g$ for 10 min to separate the precipitates for DNA extraction. Total genomic DNA from each sample was extracted using a TIANamp Bacteria DNA Kit (TIANGEN Biotech, Co., Ltd, Beijing, China) according to the manufacturer's instructions. Specific primers (341F: 5′-CCTACGGGNGGCWGCAG-3′, 805R:

5′-GACTACHVGGGTATCTAATCC-3′) were then used to amplify the V3–V4 region of the 16S rRNA gene. The reaction mixture (total volume of 30 µL) consisted of 2 µL of microbial DNA (10 ng/µL), 1 µL of each primer, 15 µL of 2× KAPA HiFi Hot Start Ready Mix, and 11 µL of sterile, deionized water. PCR was performed using a thermal cycler (Applied Biosystems 9700, USA) with the following program: 1 cycle of 3 min at 95℃; 5 cycles of 30 s at 95℃, 30 s at 45℃, and 30 s at 72℃; followed by 20 cycles of 30 s at 95℃, 30 s at 55℃, and 30 s at 72℃, concluding with a final cycle of 5 min at 72℃. PCR products were verified by electrophoresis on 1% (wt/vol) agarose gels in TBE buffer (Tris, boric acid, EDTA) and subsequently stained with ethidium bromide for visualization under UV light.

## Amplicon library construction and sequencing

AMPure XP beads were used to purify the amplicon products, and sequencing libraries were constructed by a TruSeq DNA PCR-free sample preparation kit (Illumina, USA) following the manufacturer's recommendations, with index codes added. The DNA concentration of each PCR product was measured using a Qubit 2.0 Green double-stranded DNA assay prior to sequencing, and quality control was performed using a bioanalyzer (Agilent 2100, USA). The purified amplicons were then pooled in equimolar ratios based on their concentrations. Sequencing was conducted using the Illumina MiSeq system (Illumina MiSeq, USA), according to the standard Illumina sequencing protocol. MiSeq sequencing and library construction were performed by technical staff at Sangon BioTech (Shanghai). The original data were uploaded to the NCBI SRA database (serial number: PRJNA1172367).

## Bioinformatics analysis

We tested a minimum of 60,000 reads for each individual sample to ensure that the sequencing depth is sufficient to obtain adequate coverage and rich microbial community information. Moreover, the library construction and sequencing input for each sample in the early stage were consistent. After sequencing, raw data were subjected to filtration by removing adapters, low-quality reads, and sequences longer than 480 base pairs. The filtered reads were assembled into contigs using PEAR software version 0.9.8 based on overlap. The processed fastq files generated individual fasta and qual files for further analysis. Effective tags were clustered into OTUs with ≥97% similarity using Usearch software version 11.0.667 (65). Chimeric sequences and singleton OTUs were removed, and the remaining sequences were sorted into each sample based on the OTUs. The tag sequence with the highest abundance was selected as a representative sequence within each cluster. Taxonomic classification of bacterial OTU representative sequences was performed by blasting against the RDP Database (https://bio.tools/rdp). The abundance and diversity of the samples were evaluated using the Shannon and Chao1 indices. To rigorously eliminate the potential for artifactual conclusions, all samples were rarefied to an even depth of 19,725 sequences per sample (the minimum depth observed in the CAP group) for all subsequent diversity analyses. The beta-diversity analysis was calculated using Bray–Curtis and weighted UniFrac distances and visualized by PCoA. LEfSe was employed to explore significant differences among groups and identify potential biomarkers (66). A random forest classification model was implemented using the random forest package (version 4.7–1.1) in R. The input features for the model were the relative abundances of bacterial genera, and the response variable was the disease group classification. The model was trained with the following parameters: the number of trees (ntree) was set to 500, and the number of features tried at each split (mtry) was set to the default value (the square root of the total number of features, *P*). The model's performance was rigorously evaluated using the out-of-bag error estimation. Furthermore, the predictive ability of key discriminatory genera was assessed using ROC curve analysis, with the AUC serving as the performance metric.

## Statistical analysis

Independent $t$-tests or Mann-Whitney $U$-tests were applied to continuous variables, while $\chi^2$ tests or Fisher's exact tests were used for categorical variables to compare clinical data. Statistical analyses were performed using Graphpad Prism v9.5.1, with $P < 0.05$ considered statistically significant.

## ACKNOWLEDGMENTS

This work was supported by a grant from the Wuhan Municipal Health Commission (Project no. WX21Q42).

## AUTHOR AFFILIATIONS

[1]Department of Medical Laboratory, The Central Hospital of Wuhan, Tongji Medical College, Huazhong University of Science and Technology, Wuhan, China
[2]School of Laboratory Medicine, Hubei University of Chinese Medicine, Wuhan, China
[3]Department of Clinical Laboratory, Xianning Central Hospital, The First Affiliated Hospital of Hubei University of Science and Technology, Xianning, China
[4]Hubei Provincial Engineering Research Center of Intestinal Microecological Diagnostics, Therapeutics, and Clinical Translation, Wuhan, China
[5]Hubei Center for Clinical Laboratory, Wuhan, China

## AUTHOR ORCIDs

Yingmiao Zhang  http://orcid.org/0009-0008-8798-6760
Zhongxin Lu  http://orcid.org/0000-0002-3365-0881

## FUNDING

| Funder | Grant(s) | Author(s) |
| --- | --- | --- |
| Wuhan Municipal Health Commission | WX21Q42 | Yingmiao Zhang |

## AUTHOR CONTRIBUTIONS

Jia Xu, Conceptualization, Data curation, Investigation, Methodology, Writing – original draft | Yingmiao Zhang, Conceptualization, Data curation, Funding acquisition, Investigation, Methodology, Writing – original draft | Lifeng Shi, Data curation, Investigation, Methodology, Resources | Hui Wang, Data curation, Investigation, Methodology, Resources | Ming Zeng, Data curation, Formal analysis, Resources, Validation | Zhongxin Lu, Conceptualization, Project administration, Supervision, Validation, Writing – review and editing

## DATA AVAILABILITY

The data presented in the study are deposited in the Sequence Read Archive (https://www.ncbi.nlm.nih.gov/), accession number PRJNA1172367.

## ETHICS APPROVAL

The study was approved by the Medical Ethics Committee of the Central Hospital of Wuhan, Tongji Medical College, Huazhong University of Science and Technology (Ethical No: WHZXKYL2024-158).

## ADDITIONAL FILES

The following material is available online.

## Supplemental Material

**Supplemental figures and tables (Spectrum00450-25-s0001.pdf).** Figures S1 to S4, and Tables S1 and S2.

## Open Peer Review

**PEER REVIEW HISTORY (review-history.pdf).** An accounting of the reviewer comments and feedback.

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
