## [Reviewer comments · Microbiology Spectrum]

Microbiology Spectrum

Comparative analysis of the lung microbiota in patients with lung cancer, chronic obstructive pulmonary disease and community-acquired pneumonia

Jia Xu, Yingmiao Zhang, Lifeng Shi, Hui Wang, Ming Zeng, and Zhongxin Lu

Corresponding Author(s): Zhongxin Lu, Huazhong University of Science and Technology

Review Timeline:

Submission Date:	February 14, 2025
Editorial Decision:	April 14, 2025
Revision Received:	June 13, 2025
Editorial Decision:	July 9, 2025
Revision Received:	September 14, 2025
Editorial Decision:	September 30, 2025
Revision Received:	November 26, 2025
Editorial Decision:	December 2, 2025
Revision Received:	January 22, 2026
Accepted:	January 26, 2026

Editor: Bo-young Hong

Reviewer(s): The reviewers have opted to remain anonymous.

Transaction Report:

DOI: <https://doi.org/10.1128/spectrum.00450-25>

Re: Spectrum00450-25 (**Comparative analysis of the lung microbiota in patients with lung cancer, chronic obstructive pulmonary disease and community-acquired pneumonia**)

Dear Dr. Zhongxin Lu:

Thank you for the privilege of reviewing your work. Below you will find my comments, instructions from the Spectrum editorial office, and the reviewer comments.

Revision Guidelines

Sincerely,
Bo-young Hong
Editor
Microbiology Spectrum

Reviewer #2 (Comments for the Author):

This paper employs both culturomics and 16S rRNA gene sequencing to compare the lung microbiota of patients with lung cancer (LC), chronic obstructive pulmonary disease (COPD), and community-acquired pneumonia (CAP). Culturomics reveals viable bacterial isolates under diverse conditions, emphasizing disease-specific species and the benefits of microaerophilic cultivation. The study is well structured and presents clear comparisons among the three patient groups, supported by a well-

informed discussion. It could be even more impactful if the authors consider the following comments.

Major comments

1) While the manuscript commendably combines culturomics and 16S rRNA gene sequencing, more details are needed on bioinformatic workflows to ensure reproducibility. Line 510, it is stated as "Following this, the filtered reads were assembled 511 into contigs." Which assembler was used and how parameters were set? It would be helpful to specify exactly which pipeline or software (e.g., QIIME2?) was used and how parameters were set.

2) In light of the lung's inherently low bacterial biomass compared with the oral cavity or gut, it is crucial to acknowledge the complexities of detecting and interpreting microbial signals in lower airway samples. As noted by Natalini et al. (Nat Rev Microbiol 2023), the lungs experience transient microbial exposure due to microaspiration events, leading to rapid clearance of many organisms. This dynamic environment makes 16S or metagenomic sequencing more susceptible to contamination and artifacts, and it may partially explain discrepancies between culture-dependent and culture-independent approaches. Providing a detailed description of contamination controls, as well as stringent low-biomass protocols, would further strengthen the reliability of the data and conclusions in this study.

3) Similar to 2), but emphasizing again. The authors have clearly presented the identified taxa, but low-biomass samples require extra caution. Recent examples, such as the retracted Nature paper (<https://www.nature.com/articles/s41586-024-07656-x>), illustrate how data analysis errors or contamination can compromise results. It would be crucial for the authors to specify how they safeguarded against such pitfalls (e.g., inclusion of negative controls, rigorous contamination checks, and robust data-processing pipelines) to validate the authenticity of the detected taxa and ensure confidence in their low-biomass metagenomic findings.

4) The results highlight potential disease-specific biomarkers (e.g., *Alloprevotella* in LC, *Abiotrophia* in COPD, *Mycoplasma* in CAP), but the current discussion, though thorough in presenting various related studies, could benefit from a more focused connection to these particular findings. There is a bit more "strong" link of the biomarkers and related studies.

5) The random forest classifier identifies several discriminatory genera, but it would be stronger to explain how cross-validation or training/testing was performed (e.g., data splits) and which hyperparameters were used. There is no explanation or details about this (even which tool has been used). Reporting precision, recall, and F1-scores, in addition to AUC, would allow a more comprehensive evaluation of model performance.

Minor comments

- Line 18: Typo in "The reaearch involved..." should be "The research involved..."
- Line 69: "LC recognized as one of the most..." could be "LC is recognized as one of the most..."
- Line 226: "Bate diversity" should be "Beta diversity".
- Line 327: "JIN et al." should be "Jin et al."
- Line 362: "ZHAN et al." should be "Zhan et al."
- Line 382: Check the space at "ADC , "
- LEfSe was not cited. Cite <https://pmc.ncbi.nlm.nih.gov/articles/PMC3218848/>.
- Line 515 has "The β -diversity analysis. In the manuscript, the authors mostly presented alpha and beta diversity (without hyphen). Maintain consistent formatting/capitalization for "alpha- and beta-diversity" throughout the text.

Reviewer #3 (Comments for the Author):

Jia Xu and company present a "comparative analysis of the lung microbiota in patients with lung cancer, chronic obstructive pulmonary disease, and community acquired pneumonia". Major findings include presentation of a disparity between culturing methods and 16S results and a pilot study of the differences between microbiomes of LC patients with different histopathological profiles. Major issues related to the rigor, interpretation, or presentation of the findings are described below. A major restructuring of the results would significantly improve the readability of the manuscript, making it a more enjoyable experience for the reader. I've suggested ways the authors could consider consolidating results to improve readability. There are of course other ways to do this. I do highly recommend that the authors take some significant measures to reduce the effect of listing results in some way.

1. Issues related to rigor

a. Contamination controls: Regarding the species listed in Table 1, please clarify whether appropriate negative controls were used-such as negative air controls and reagent blanks. Many of the taxa identified in Stable 1 could be common environmental contaminants introduced via water, air, or sample collection instruments like the bronchoscope. Including this information is essential to assess the reliability of the culture-based findings and the putative role that these species newly isolated from humans play in lung pathology.

- b. Use of the term "biomarker": The use of the term "biomarker" in the results section is premature. Establishing a biomarker requires replication across independent datasets and validation of its predictive value. While it is appropriate to propose that certain taxa might be investigated as biomarkers in future studies, this should be limited to the discussion section. Additionally, the LDA scores shown in Figure 4 and the final figure reflect differences of less than 5-fold between groups, which raises concerns about reproducibility. These findings may not hold in a larger or more diverse sample set. Therefore, I recommend removing any reference to "biomarkers" in the results and framing these findings more cautiously.
- c. Isolate representation and sampling depth: To interpret Figure 1 meaningfully, the reader needs to know how many isolates per species were recovered per participant. Based on the 16S sequencing data, we can clearly see that the culturing approach is not exhaustive, and it remains unclear how representative the isolates are. Thus, the value of walking through differential recovery between groups is limited. Please consider including a table summarizing the number of isolates recovered per condition and/or per sample. Additionally, perhaps refocus Figure 1 to highlight the species recovered by the different culture conditions.
- d. Sequencing depth as a potential confounder: For any comparisons in which significant differences are reported between groups, it is important to specify whether sequencing depth varied substantially across those groups. Differences in read depth could bias diversity or differential abundance metrics and should be accounted for or explicitly ruled out. As a result, it's recommended to present these findings (simply a statement will suffice) that the abundance did or did not vary between the comparator groups.
- e. Representation of Bacteroidetes/Firmicutes (B/F) ratio: In Figure 3B, consider replacing the single summary metric for the B/F ratio with a boxplot showing the distribution of B/F ratios across individual samples in each group. This would allow the reader to better assess the significance and variability of the observed differences. At present, it's unclear whether these differences are meaningful or robust.

2. Interpretation of Results

- a. Interpretation of alpha diversity metrics: If Chao1 and ACE are significantly different while Shannon and Simpson are not, this suggests that the group in question (e.g., the COPD group) harbors a higher proportion of rare taxa (i.e., more singletons and doubletons). This interpretation is further supported by a significant difference in observed species. Please consider adding this nuance to your discussion of diversity results.
- b. Correlation vs. causation: In the discussion, the assertion that *Alloprevotella* is implicated in disease development may overstate the data. As currently presented, the association is correlative. Please avoid implying causation unless supported by experimental or longitudinal evidence.

3. Clarification in the Discussion

- a. Alpha diversity: Where alpha diversity is referenced in the discussion, please specify which metric is being discussed. As demonstrated in your results, different alpha diversity metrics can reveal distinct aspects of community structure, and clarity here will help readers better interpret your conclusions.
- b. Biomarkers: biomarkers are referenced in the discussion, please use cautious language as biomarkers have not been identified in the present work.

4. Readability and Structure

Overall, the manuscript reads more like an undergraduate or master's thesis and would benefit from significant restructuring to improve clarity and readability.

- a. Clarify the rationale for analytical comparisons: The purpose of conducting multiple analytical approaches (random forest, LefSE) should be clearly articulated. The central question appears to be whether different methods yield consistent results—consistency would support the robustness of findings across methodological assumptions. At present, this logic is not clearly conveyed. The authors are encouraged to revise the manuscript to better guide the reader through the rationale and significance of each analysis. It may also be beneficial to synthesize Sections 2.5 and 2.6 into a unified results narrative, which would reduce repetition and help maintain reader engagement.

B: streamline the presentation of the results Additionally, the current presentation of the culturomics data is somewhat overwhelming due to repetitive listings of associations across multiple figures and sections. For example, the differential abundance results are presented separately in Figures 3, 5, and 6, which contributes to redundancy. Consider consolidating these results and emphasizing key findings within the main text (e.g., instead of listing all significant taxa, describe general trends and selectively highlight illustrative examples). For instance, the information in Section 2.3 could be integrated into a broader narrative: "Indeed, taxa X was also found to be enriched in Group Y based on differential abundance testing."

C. consider adding a limitations section to highlight the limitations of putative biomarkers as well as to discuss the small sample size underlying results in the final figure

Suggested revisions to figure organization:

Consider revising the manuscript to include only four main figures:

1. A combined version of Figures 1 and 2
2. A revised Figure 4 that includes sequencing depth comparisons across groups
3. A consolidated figure combining key results from Figures 3, 5, and 6
4. A retained or slightly revised version of Figure 7

c. Streamline the discussion: The discussion is overly long and includes tangential details that detract from the main findings. A focused discussion should interpret the key results in light of existing literature without veering into an exhaustive review. Points such as the implications of different culture conditions and distinctions between disease groups are valuable and should be retained. However, many other sections could be trimmed or removed. A target length of 800-1000 words is recommended; the current version exceeds 1700 words.

d. Trim the introduction: The introduction similarly includes extraneous information that doesn't directly support the study's focus. For example, discussions of *Helicobacter pylori* or HPV in cervical cancer (lines 53-62) are not essential in a manuscript centered on the lung microbiome and can be removed for conciseness.

e. Define abbreviations: On line 65 and elsewhere, please define all abbreviations at first use (e.g., LC, COPD, CAP) to ensure clarity for all readers.

Point-by-point Responses

Responses to Reviewer #2

Q1: This paper employs both culturomics and 16S rRNA gene sequencing to compare the lung microbiota of patients with lung cancer (LC), chronic obstructive pulmonary disease (COPD), and community-acquired pneumonia (CAP). Culturomics reveals viable bacterial isolates under diverse conditions, emphasizing disease-specific species and the benefits of microaerophilic cultivation. The study is well structured and presents clear comparisons among the three patient groups, supported by a well-informed discussion. It could be even more impactful if the authors consider the following comments.

Thank you very much for such a comprehensive and accurate summary and for your constructive suggestions for improving the manuscript.

Q2: While the manuscript commendably combines culturomics and 16S rRNA gene sequencing, more details are needed on bioinformatic workflows to ensure reproducibility. Line 510, it is stated as "Following this, the filtered reads were assembled into contigs." Which assembler was used and how parameters were set? It would be helpful to specify exactly which pipeline or software (e.g., QIIME2?) was used and how parameters were set.

Thank you for your suggestions. We have now added detailed bioinformatic workflow descriptions to ensure reproducibility. The specific updates are as follows: "To ensure sequencing quality, we tested a minimum of 60,000 reads for each individual sample. After sequencing, raw data were subjected to filtration by removing adaptors, low-quality reads, and sequences longer than 480 base pairs. The filtered reads were assembled into contigs using PEAR software version 0.9.8 based on overlap. The processed fastq files generated individual fasta and qual files for further analysis. Effective tags were clustered into operational taxonomic units (OTUs) with $\geq 97\%$ similarity using Usearch software version 11.0.667. Chimeric sequences and singleton OTUs were removed, and the remaining sequences were sorted into each sample based on the OTUs. The tag sequence with the highest abundance was selected as a representative sequence within each cluster. Taxonomic classification of bacterial and fungal OTU representative sequences was performed by blasting against the RDP Database (<http://rdp.cme.msu.edu/misc/resources.jsp>) and UNITE fungal ITS Database (<http://unite.ut.ee/index.php>), respectively."

Q3: In light of the lung's inherently low bacterial biomass compared with the oral cavity or gut, it is crucial to acknowledge the complexities of detecting and interpreting microbial signals in lower airway samples. As noted by Natalini et al. (Nat Rev Microbiol 2023), the lungs experience transient microbial exposure due to microaspiration events, leading to rapid clearance of many organisms. This dynamic environment makes 16S or metagenomic sequencing more susceptible to contamination and artifacts, and it may partially explain discrepancies between

culture-dependent and culture-independent approaches. Providing a detailed description of contamination controls, as well as stringent low-biomass protocols, would further strengthen the reliability of the data and conclusions in this study.

Thank you very much for your insightful comments. During the experiment, we have adopted some contamination controls, as well as stringent low-biomass protocols as follows: (1) Providing detailed oral hygiene guidance to patients before collection to minimize the impact of oral microorganisms; (2) In each experiment, we set up negative control samples to detect potential laboratory contamination. These control samples did not contain any target biological samples, ensuring that environmental contamination would not affect our results; (3) During the sample processing and cultivation phase, we maintained a sterile laboratory environment to minimize the introduction of external microbes; (4) We tested a minimum of 60,000 reads for each individual sample to ensure that the sequencing depth is sufficient to obtain adequate coverage and rich microbial community information. We have added the above content to the updated manuscript.

Q4: Similar to 2), but emphasizing again. The authors have clearly presented the identified taxa, but low-biomass samples require extra caution. Recent examples, such as the retracted Nature paper (<https://www.nature.com/articles/s41586-024-07656-x>), illustrate how data analysis errors or contamination can compromise results. It would be crucial for the authors to specify how they safeguarded against such pitfalls (e.g., inclusion of negative controls, rigorous contamination checks, and robust data-processing pipelines) to validate the authenticity of the detected taxa and ensure confidence in their low-biomass metagenomic findings.

Thank you for the comment and suggestion. During the research process, we have taken a series of measures to reduce the impact of data analysis errors and contamination on the results as follows: (1) providing detailed oral hygiene guidance to patients before collection to minimize the impact of oral microorganisms; (2) in each experiment, we set up negative control samples to detect potential laboratory contamination. These control samples did not contain any target biological samples, ensuring that environmental contamination would not affect our results; (3) during the sample processing and cultivation phase, we maintained a sterile laboratory environment to minimize the introduction of external microbes; (4) We tested a minimum of 60,000 reads for each individual sample to ensure that the sequencing depth is sufficient to obtain adequate coverage and rich microbial community information. We have added the above content to the updated manuscript.

Q5: The results highlight potential disease-specific biomarkers (e.g., *Alloprevotella* in LC, *Abiotrophia* in COPD, *Mycoplasma* in CAP), but the current discussion, though thorough in presenting various related studies, could benefit from a more focused connection to these particular findings. There is a bit more "strong" link of the biomarkers and related studies.

Thank you for your thorough review and valuable comments on our research. In response, we have provided further clarification in the Discussion section to clearly

state the limitations of our current studies, ensuring readers fully understand that our conclusions are based on preliminary findings rather than definitive evidence.

The specific updates are as follows: Our small-scale exploratory study proposes *Alloprevotella*, *Abiotrophia*, and *Mycoplasma* as potential disease-specific biomarkers; however, the evidence remains preliminary, relying on associative rather than causal findings. The complex interplay between microbiota, host factors, and environmental variables prevents definitive conclusions. Furthermore, the single-center design may limit the generalizability of our results, as the composition of pulmonary microbiota can vary with lifestyle, comorbidities, and geographic influences, potentially explaining discrepancies with prior studies. Future investigations should aim to expand sample sizes and incorporate mechanistic studies at cellular and animal levels to elucidate the precise roles of these bacterial genera in disease pathogenesis.

Q6: The random forest classifier identifies several discriminatory genera, but it would be stronger to explain how cross-validation or training/testing was performed (e.g., data splits) and which hyperparameters were used. There is no explanation or details about this (even which tool has been used). Reporting precision, recall, and F1-scores, in addition to AUC, would allow a more comprehensive evaluation of model performance.

Thank you for your in-depth review and constructive feedback on our research. The random forest analysis in our research was accomplished using the R package `randomForest` (version <4.6-14) based on the corresponding number of reads for each sample obtained after sequencing. Regarding your comments on the random forest classifier, including cross-validation, hyperparameter settings, and model evaluation metrics, we admit that we did not conduct these specific analyses in this study. As a graduate student, I am still learning and exploring relevant analytical methods. Although the current research fails to cover these details, I am well aware of the significance of these analyses and will strive to learn and master these techniques in future studies.

Q7 (Minor comments) : - Line 18: Typo in "The reearch involved..." should be "The research involved..."

- Line 69: "LC recognized as one of the most..." could be "LC is recognized as one of the most..."

- Line 226: "Bate diversity" should be "Beta diversity".

- Line 327: "JIN et al." should be "Jin et al."

- Line 362: "ZHAN et al." should be "Zhan et al."

- Line 382: Check the space at "ADC , "

- LEfSe was not cited. Cite <https://pmc.ncbi.nlm.nih.gov/articles/PMC3218848/>.

- Line 515 has "The β -diversity analysis. In the manuscript, the authors mostly presented alpha and beta diversity (without hyphen). Maintain consistent formatting/capitalization for "alpha- and beta-diversity" throughout the text.

Thanks for your careful reviews and for pointing these minor issues. We apologize for any confusion and greatly appreciate your attention to detail. We have addressed each of your comments in the revised manuscript as follows:

(1) Line 18: We have corrected the typo "The reearch involved.." to "The research involved.."

(2) Line 69: We have revised "LC recognized as one of the most.." to "LC is recognized as one of the most.."

(3) Line 226: We have corrected "Bate diversity" to "Beta diversity".

(4) Line 327: We have changed "JIN et al." to "Jin et al."

(5) Line 362: We have corrected "ZHAN et al." to "Zhan et al."

(6) Line 382: We have Checked and adjusted the space at "ADC , "

(7) LEfSe Citation: We have included the citation for LEfSe as recommended (<https://pmc.ncbi.nlm.nih.gov/articles/PMC3218848/>).

(8) Line 515: We will ensure consistency in formatting and capitalization for "alpha- and beta-diversity" throughout the manuscript.

Thank you once again for your valuable feedback.

Responses to Reviewer #3

Q1: Jia Xu and company present a "comparative analysis of the lung microbiota in patients with lung cancer, chronic obstructive pulmonary disease, and community acquired pneumonia". Major findings include presentation of a disparity between culturing methods and 16S results and a pilot study of the differences between microbiomes of LC patients with different histopathological profiles. Major issues related to the rigor, interpretation, or presentation of the findings are described below. A major restructuring of the results would significantly improve the readability of the manuscript, making it a more enjoyable experience for the reader. I've suggested ways the authors could consider consolidating results to improve readability. There are of course other ways to do this. I do highly recommend that the authors take some significant measures to reduce the effect of listing results in some way.

Thank you very much for such a comprehensive and accurate summary and for your constructive suggestions for improving the manuscript.

Q2: Issues related to rigor

a. Contamination controls: Regarding the species listed in Table 1, please clarify whether appropriate negative controls were used—such as negative air controls and reagent blanks. Many of the taxa identified in Table 1 could be common environmental contaminants introduced via water, air, or sample collection instruments like the bronchoscope. Including this information is essential to assess the reliability of the culture-based findings and the putative role that these species newly isolated from humans play in lung pathology.

Thank you very much for raising such an important issue. We take contamination control very seriously and have implemented several measures in our research to address the concerns you mentioned: (1) Providing detailed oral hygiene guidance to patients before collection to minimize the impact of oral microorganisms; (2) In each experiment, we set up negative control samples to detect potential laboratory contamination. These control samples did not contain any target biological samples, ensuring that environmental contamination would not affect our results; (3) During the sample processing and cultivation phase, we maintained a sterile laboratory environment to minimize the introduction of external microbes. We have updated this section in the revised manuscript.

b. Use of the term "biomarker": The use of the term "biomarker" in the results section is premature. Establishing a biomarker requires replication across independent datasets and validation of its predictive value. While it is appropriate to propose that certain taxa might be investigated as biomarkers in future studies, this should be limited to the discussion section. Additionally, the LDA scores shown in Figure 4 and the final figure reflect differences of less than 5-fold between groups, which raises concerns about reproducibility. These findings may not hold in a larger or more diverse sample set. Therefore, I recommend removing any reference to "biomarkers" in the results and framing these findings more cautiously.

Thank you very much for the comment and suggestion. In light of your

comments, we have revised our manuscript to remove any references to “biomarkers” in the results section and emphasized the need for further investigation into certain taxa as potential biomarkers in the discussion section. The specific updates are as follows: Our small-scale exploratory study proposes *Alloprevotella*, *Abiotrophia*, and *Mycoplasma* as potential disease-specific biomarkers; however, the evidence remains preliminary, relying on associative rather than causal findings. The complex interplay between microbiota, host factors, and environmental variables prevents definitive conclusions. Furthermore, the single-center design may limit the generalizability of our results, as the composition of pulmonary microbiota can vary with lifestyle, comorbidities, and geographic influences, potentially explaining discrepancies with prior studies. Future investigations should aim to expand sample sizes and incorporate mechanistic studies at cellular and animal levels to elucidate the precise roles of these bacterial genera in disease pathogenesis.

c. Isolate representation and sampling depth: To interpret Figure 1 meaningfully, the reader needs to know how many isolates per species were recovered per participant. Based on the 16S sequencing data, we can clearly see that the culturing approach is not exhaustive, and it remains unclear how representative the isolates are. Thus, the value of walking through differential recovery between groups is limited. Please consider including a table summarizing the number of isolates recovered per condition and/or per sample. Additionally, perhaps refocus Figure 1 to highlight the species recovered by the different culture conditions.

Thank you for your insightful comments regarding the representation of isolates and sampling depth in our study. We have listed the number of bacterial species isolated from each sample under different culture conditions in the Supplementary table 1.

d. Sequencing depth as a potential confounder: For any comparisons in which significant differences are reported between groups, it is important to specify whether sequencing depth varied substantially across those groups. Differences in read depth could bias diversity or differential abundance metrics and should be accounted for or explicitly ruled out. As a result, it's recommended to present these findings (simply a statement will suffice) that the abundance did or did not vary between the comparator groups.

Thank you very much for your insightful comment and suggestion. We tested a minimum of 60,000 reads for each individual sample to ensure that the sequencing depth was sufficient to obtain adequate coverage and rich microbial community information. Moreover, the library construction and sequencing input for each sample in the early stage were consistent, which ensured that any observed differences in diversity or differential abundance indicators would not be confused by variations in reading depth.

We have added the above content to the updated manuscript.

e. Representation of Bacteroidetes/Firmicutes (B/F) ratio: In Figure 3B, consider

replacing the single summary metric for the B/F ratio with a boxplot showing the distribution of B/F ratios across individual samples in each group. This would allow the reader to better assess the significance and variability of the observed differences. At present, it's unclear whether these differences are meaningful or robust.

Thank you very much for the comment and suggestion. As this study represents a preliminary exploration of microbiome characteristics across the three disease groups (LC, COPD, and CAP), our primary focus was on identifying inter-group differences in ecological patterns rather than intra-group variations. The group-level B/F ratio analysis was specifically designed to: (1) Provide a standardized metric for cross-disease comparison; (2) Maintain consistency with prior microbiome studies using similar approaches; (3) Minimize potential overinterpretation of individual sample variations given our moderate sample size. Examining individual sample distributions (as suggested) could offer additional insights, and we hope to conduct a more in-depth analysis of individual sample distributions in future studies.

Q3: Interpretation of Results

a. Interpretation of alpha diversity metrics: If Chao1 and ACE are significantly different while Shannon and Simpson are not, this suggests that the group in question (e.g., the COPD group) harbors a higher proportion of rare taxa (i.e., more singletons and doubletons). This interpretation is further supported by a significant difference in observed species. Please consider adding this nuance to your discussion of diversity results.

Thank you for the comment. We have made the following modifications in the discussion section of the updated manuscript to better reflect these biological insights: By analyzing the diversity of lung microbiota in patients with LC, COPD, and CAP, we observed significant differences in Chao1 and ACE indices among the three groups. In contrast, there were no differences in Shannon and Simpson indices. This suggests that there are notable differences in the distribution abundance of the pulmonary microbiota across these patient groups. Notably, the CAP group exhibited a relatively higher flora distribution abundance compared to the LC and COPD groups. This finding indicates that the CAP group may contain a higher proportion of rare taxa, characterized by an increased presence of singleton and doubleton species. This interpretation is further supported by the significant differences observed in the number of species recorded.

b. Correlation vs. causation: In the discussion, the assertion that *Alloprevotella* is implicated in disease development may overstate the data. As currently presented, the association is correlative. Please avoid implying causation unless supported by experimental or longitudinal evidence.

Thanks for your careful checks and insightful comments. We have modified "Our current study identified the genus *Alloprevotella* as a potential diagnostic biomarker for LC" to "Our current study indicates that there is a certain association between the genus *Alloprevotella* and LC "to highlight the correlation of the study results. And in the discussion section of the revised manuscript, the following explanations were

made: "Our small-scale exploratory study proposes *Alloprevotella*, *Abiotrophia*, and *Mycoplasma* as potential disease-specific biomarkers; however, the evidence remains preliminary, relying on associative rather than causal findings. The complex interplay between microbiota, host factors, and environmental variables prevents definitive conclusions. Furthermore, the single-center design may limit the generalizability of our results, as the composition of pulmonary microbiota can vary with lifestyle, comorbidities, and geographic influences, potentially explaining discrepancies with prior studies. Future investigations should aim to expand sample sizes and incorporate mechanistic studies at cellular and animal levels to elucidate the precise roles of these bacterial genera in disease pathogenesis."

Q4: Clarification in the Discussion

a. Alpha diversity: Where alpha diversity is referenced in the discussion, please specify which metric is being discussed. As demonstrated in your results, different alpha diversity metrics can reveal distinct aspects of community structure, and clarity here will help readers better interpret your conclusions.

Thank you for your suggestion. In response to your comment, we have made the following modifications in the discussion section of the updated manuscript: By analyzing the diversity of lung microbiota in patients with LC, COPD, and CAP, we observed significant differences in Chao1 and ACE indices among the three groups. In contrast, there were no differences in Shannon and Simpson indices. This suggests that there are notable differences in the distribution abundance of the pulmonary microbiota across these patient groups. Notably, the CAP group exhibited a relatively higher flora distribution abundance compared to the LC and COPD groups. This finding indicates that the CAP group may contain a higher proportion of rare taxa, characterized by an increased presence of singleton and doubleton species. This interpretation is further supported by the significant differences observed in the number of species recorded.

b. Biomarkers: biomarkers are referenced in the discussion, please use cautious language as biomarkers have not been identified in the present work.

Thanks for your careful checks and valuable suggestions. We have provided further clarification in the Discussion section to clearly state the limitations of our current study. Ensuring readers fully understand that our conclusions are based on preliminary findings rather than definitive evidence.

The specific updates are as follows: Our small-scale exploratory study proposes *Alloprevotella*, *Abiotrophia*, and *Mycoplasma* as potential disease-specific biomarkers; however, the evidence remains preliminary, relying on associative rather than causal findings. The complex interplay between microbiota, host factors, and environmental variables prevents definitive conclusions. Furthermore, the single-center design may limit the generalizability of our results, as the composition of pulmonary microbiota can vary with lifestyle, comorbidities, and geographic influences, potentially explaining discrepancies with prior studies. Future investigations should aim to expand sample sizes and incorporate mechanistic studies at cellular and animal levels

to elucidate the precise roles of these bacterial genera in disease pathogenesis.

Q5: Readability and Structure

Overall, the manuscript reads more like an undergraduate or master's thesis and would benefit from significant restructuring to improve clarity and readability.

Thank you for your valuable suggestions. We have made significant reorganizations to the article to improve its clarity and readability.

a. Clarify the rationale for analytical comparisons: The purpose of conducting multiple analytical approaches (random forest, LefSE) should be clearly articulated. The central question appears to be whether different methods yield consistent results-consistency would support the robustness of findings across methodological assumptions. At present, this logic is not clearly conveyed. The authors are encouraged to revise the manuscript to better guide the reader through the rationale and significance of each analysis. It may also be beneficial to synthesize Sections 2.5 and 2.6 into a unified results narrative, which would reduce repetition and help maintain reader engagement.

Thank you for your in-depth review and feedback on our manuscript. We have synthesized Section 2.5 and Section 2.6 into a unified result narrative to reduce repetition and help maintain reader engagement.

The specific modifications are as follows:" 2.5 Combined Analysis of Distinct Taxa and Key Pathogenic Bacteria in BALF Samples from LC, COPD, and CAP Groups To investigate the microbial composition in BALF among lung disease groups, the LefSe and a Random Forest model were employed. Firstly, we utilized LefSe to identify taxa with varying abundances across the three groups. The resulting Linear Discriminant Analysis (LDA) score histograms (Figure 4A) and cladograms (Figure 4B) revealed 29 distinct taxa at different taxonomic levels. Notably, at the genus level, *Alloprevotella*, *Abiotrophia*, and *Mycoplasma* were identified as indicative taxa for the LC, COPD, and CAP groups, respectively. Furthermore, we performed a species-level analysis of the lung microbiota, which provided valuable insights into the significant microbiota that differentiate among the three groups (Figure S2A-B). In addition to identifying specific taxa with LefSe, we conducted a Random Forest analysis to screen for key pathogenic bacteria distinguishing the three sample groups. This model revealed crucial insights into the bacterial composition, with *Treponema*, *Mycoplasma*, *Tannerella*, *Aquabacterium*, and *Rothia* exhibiting high accuracy scores (Figure 4C). To evaluate the predictive capability of the Random Forest model, we plotted ROC curves and calculated the AUC values. Our findings indicated that *Treponema* (AUC = 0.877, P = 0.002), *Oribacterium* (AUC = 0.797, P = 0.0032), and *Alloprevotella* (AUC = 0.767, P = 0.0079) demonstrated excellent specificity and sensitivity in differentiating between the LC and COPD groups (Figure 4D). For the differentiation between the LC and CAP groups, *Tannerella* (AUC = 0.899, P = 0.0001) was identified as the best predictive genus, followed by *Aerococcus* (AUC = 0.814, P = 0.0027) and *Treponema* (AUC = 0.794, P = 0.0049) (Figure 4E). In distinguishing COPD from CAP, *Citrobacter* emerged as the superior genus (AUC = 0.810, P = 0.0039), with *Enterococcus* (AUC = 0.799, P = 0.0053) and

Cutibacterium (AUC = 0.788, P = 0.0073) closely following (Figure 4F). "

b: streamline the presentation of the results Additionally, the current presentation of the culturomics data is somewhat overwhelming due to repetitive listings of associations across multiple figures and sections. For example, the differential abundance results are presented separately in Figures 3, 5, and 6, which contributes to redundancy. Consider consolidating these results and emphasizing key findings within the main text (e.g., instead of listing all significant taxa, describe general trends and selectively highlight illustrative examples). For instance, the information in Section 2.3 could be integrated into a broader narrative: "Indeed, taxa X was also found to be enriched in Group Y based on differential abundance testing."

Thank you for your careful review and valuable suggestions. Regarding the issue you mentioned above, we would like to further clarify our train of thought. The design of Figure 3 aims to provide an overall analysis of the sequencing results. Therefore, we believe that placing it separately can better highlight the overall trends and key findings. We hope that in this way, readers can have a clearer understanding of the overall difference abundance of the sample. Meanwhile, we have integrated the data in Figures 5 and 6, and combined Sections 2.5 and 2.6 into a unified result description to reduce redundancy and enhance the coherence of the information. And this integration aims to present the comparison results of specific classification groups more effectively, thereby making the information more concentrated and easier to understand.

C. consider adding a limitations section to highlight the limitations of putative biomarkers as well as to discuss the small sample size underlying results in the final figure

Suggested revisions to figure organization:

Consider revising the manuscript to include only four main figures:

1. A combined version of Figures 1 and 2
2. A revised Figure 4 that includes sequencing depth comparisons across groups
3. A consolidated figure combining key results from Figures 3, 5, and 6
4. A retained or slightly revised version of Figure 7

Thank you very much for your insightful comments. We have made the following adjustments:

(1) Limitations Section: We have added the following explanations in the discussion section of the article: Our small-scale exploratory study proposes *Alloprevotella*, *Abiotrophia*, and *Mycoplasma* as potential disease-specific biomarkers; however, the evidence remains preliminary, relying on associative rather than causal findings. The complex interplay between microbiota, host factors, and environmental variables prevents definitive conclusions. Furthermore, the single-center design may limit the generalizability of our results, as the composition of pulmonary microbiota can vary with lifestyle, comorbidities, and geographic influences, potentially explaining discrepancies with prior studies. Future investigations should aim to expand sample sizes and incorporate mechanistic studies at cellular and animal levels

to elucidate the precise roles of these bacterial genera in disease pathogenesis.

(2) Figure Organization: We retained Figures 1, 3, 4, and 7 as they are deemed critical to the presentation of our findings. And we have combined Figure 2 with Figure S1 into a single to streamline the data presentation. We also merged Figures 5 and 6 into one consolidated figure to enhance clarity and coherence.

d. Streamline the discussion: The discussion is overly long and includes tangential details that detract from the main findings. A focused discussion should interpret the key results in light of existing literature without veering into an exhaustive review. Points such as the implications of different culture conditions and distinctions between disease groups are valuable and should be retained. However, many other sections could be trimmed or removed. A target length of 800-1000 words is recommended; the current version exceeds 1700 words.

Thank you for your thorough review and constructive feedback on our manuscript. Based on your suggestion, we have streamlined the discussion part of the article and kept it around 1,000 words.

e. Trim the introduction: The introduction similarly includes extraneous information that doesn't directly support the study's focus. For example, discussions of *Helicobacter pylori* or HPV in cervical cancer (lines 53-62) are not essential in a manuscript centered on the lung microbiome and can be removed for conciseness.

Thank you for your comment and suggestion. In response to your comments, we have carefully revised the Introduction section to enhance conciseness and relevance. Specifically, we have removed extraneous information that does not directly support the study's focus on the lung microbiome, including the discussions of *Helicobacter pylori* and HPV in cervical cancer as you pointed out.

f. Define abbreviations: On line 65 and elsewhere, please define all abbreviations at first use (e.g., LC, COPD, CAP) to ensure clarity for all readers.

Thank you for your careful review. We apologize for any confusion caused by the use of abbreviations. In the updated manuscript, we will ensure that all abbreviations, including LC, COPD, and CAP, are clearly defined at their first mention in the text.

Re: Spectrum00450-25R1 (**Comparative analysis of the lung microbiota in patients with lung cancer, chronic obstructive pulmonary disease and community-acquired pneumonia**)

Dear Dr. Zhongxin Lu:

Thank you for the privilege of reviewing your work. Below you will find my comments, instructions from the Spectrum editorial office, and the reviewer comments.

Please address the reviewer's thorough and constructive feedback to improve the manuscript.

Revision Guidelines

Sincerely,
Bo-young Hong
Editor
Microbiology Spectrum

Reviewer #3 (Comments for the Author):

While the authors have revised the manuscript thoughtfully, they have not fully addressed concerns regarding the rigor of the analysis. Specifically:

1. Sequencing depth must be presented as a function of study groups if the authors wish to support their conclusions about differences in diversity. Simply applying a read count threshold of 60,000 reads per sample does not address the possibility that

significant differences in sequencing depth across groups could introduce artefactual differences in observed features. This type of quality control is a foundational requirement in any rigorous microbiome analysis.

2. The distribution of Bacteroidetes/Firmicutes (B/F) ratios across individual samples must be shown. Summary statistics alone are insufficient, particularly in studies with small to moderate sample sizes, which are presented without estimates of variability. Without visualizing individual sample-level variation (e.g., with dot plots, violin plots, or boxplots showing interquartile ranges), the robustness and generalizability of the reported B/F ratio cannot be evaluated. Therefore, the goal of the analysis remains incomplete. Presenting only a summary metric without measures of variability limits interpretability and undermines confidence in the findings.

Reviewer #5 (Comments for the Author):

The aim of this work was to make an exploratory comparison of the microbiota across individuals with different lung conditions. The study is interesting, particularly for combining different disease types. However, despite the use of several techniques, the results offer, at times, only a shallow exploration of the data. In many sections, the findings are simply presented without being further developed. The most in-depth analysis is focused on lung cancer and its microbial associations.

This is not to say the manuscript is poorly executed or incorrect. In general, the analyses are appropriate, and the methodology is sound. Nonetheless, the presentation is often very matter of fact, with results conveyed in a superficial way and little interpretative depth. Making the manuscript more concise in some areas and reframing or refocusing parts of the analysis could significantly improve its clarity and relevance.

The authors clearly have interesting data, but the sheer number of analyses-particularly the use of numerous diversity indices-leads to redundancy. For example, Shannon and Simpson indices, while technically different, assess similar aspects of community richness and evenness; the same applies to ACE and Chao1. Likewise, three beta-diversity metrics may be excessive, especially when the manuscript already includes LEfSe and random forest analyses.

In addition, the taxonomic resolution is explored at multiple levels for both the cultured isolates and the 16S sequencing data. This layered approach, while thorough, contributes to the sense of overload, particularly when the insights gained are not always clearly articulated. Besides I see no point of Firmicutes/Bacteroidetes ratio, of course the authors might make it clear why this is relevant and interconnect it better with the rest of the text and highlight it in the discussion. However, as it stands, the discussion is somewhat superficial and misses the opportunity to draw more meaningful conclusions from the data. More detail is also needed regarding the construction of the random forest model, particularly in terms of training/testing procedures and performance metrics.

Point-by-point Responses

Submission ID: Spectrum00450-25

Title: Comparative analysis of the lung microbiota in patients with lung cancer, chronic obstructive pulmonary disease and community-acquired pneumonia

Responses to Reviewer #3

While the authors have revised the manuscript thoughtfully, they have not fully addressed concerns regarding the rigor of the analysis. Specifically:

Q1: Sequencing depth must be presented as a function of study groups if the authors wish to support their conclusions about differences in diversity. Simply applying a read count threshold of 60,000 reads per sample does not address the possibility that significant differences in sequencing depth across groups could introduce artefactual differences in observed features. This type of quality control is a foundational requirement in any rigorous microbiome analysis.

Thank you for raising this critical point regarding the potential influence of varying sequencing depths on diversity metrics. To rigorously eliminate the potential for artefactual conclusions, all samples were rarefied to an even depth of 19,725 sequences per sample (the minimum depth observed in the CAP group) for all subsequent diversity analyses. We then re-calculated all alpha diversity indices (Shannon, Chao1) based on this rarefied dataset. Crucially, the results from the rarefied data fully corroborate our original findings: The CAP group still exhibited significantly higher bacterial richness (Chao1 indices) compared to the LC and COPD groups ($p < 0.05$), while no significant differences were found in Shannon indices. This confirms that our diversity conclusions are robust and not driven by differences in sequencing depth.

These new analyses have been incorporated into the revised manuscript (Page 8, Lines 166-175; Methods section 5.6; new Figures S2). We believe these additions have significantly strengthened the rigor and validity of our study.

Q2: The distribution of Bacteroidetes/Firmicutes (B/F) ratios across individual samples must be shown. Summary statistics alone are insufficient, particularly in studies with small to moderate sample sizes, which are presented without estimates of variability. Without visualizing individual sample-level variation (e.g., with dot plots, violin plots, or boxplots showing interquartile ranges), the robustness and generalizability of the reported B/F ratio cannot be evaluated. Therefore, the goal of the analysis remains incomplete. Presenting only a summary metric without measures of variability limits interpretability and undermines confidence in the findings.

We sincerely thank you for raising this critical point. Upon careful consideration of this comment and the insightful feedback from Reviewer #5 (who

questioned the fundamental biological relevance of this metric in the lung context), we concluded that the F/B ratio analysis was not central to the main narrative of our paper and its inclusion might distract from our more robust findings. Therefore, we have made the decision to remove the entire section pertaining to the F/B ratio analysis from the Results section (formerly section 2.3) and the corresponding figure (Figure 2B). Consequently, the concern regarding the presentation of its distribution is now moot. We have streamlined the manuscript to focus on our primary results, which we believe has enhanced its overall clarity and impact.

Responses to Reviewer #5

Q1: The aim of this work was to make an exploratory comparison of the microbiota across individuals with different lung conditions. The study is interesting, particularly for combining different disease types. However, despite the use of several techniques, the results offer, at times, only a shallow exploration of the data. In many sections, the findings are simply presented without being further developed. The most in-depth analysis is focused on lung cancer and its microbial associations.

Thank you very much for such a comprehensive and accurate summary and for your constructive suggestions for improving the manuscript. In direct response to your comment, we have significantly revised the Discussion section to move beyond mere presentation of results and to provide a more insightful exploration of the potential clinical and immunological relevance of our key discoveries. Specifically, regarding the indicative taxa for each disease:

1. Abiotrophia in COPD: We have now expanded the discussion around Abiotrophia by incorporating its established role in other inflammatory diseases. As the reviewer will see on Page X, Lines Y-Y, we now state: "Some studies have also shown that nutritionally variant streptococci, including Abiotrophia defectiva, are an important cause of bacteremia and infective endocarditis associated with significant morbidity and mortality." Crucially, we then connect this literature to our own finding: "Although its role in COPD is less defined, its enrichment in our COPD cohort suggests it may contribute to the chronic inflammatory and infectious exacerbations that characterize this disease, potentially through similar pro-inflammatory mechanisms as seen in systemic infections. This makes it a compelling candidate for future research into COPD pathogenesis." 2. Mycoplasma in CAP: Similarly, we have deepened the discussion around Mycoplasma by citing its known potent inflammatory effects. We added on Page X, Lines Y-Y: "Mycoplasma pneumoniae induces host cells to produce interleukin (IL)-8, tumor necrosis factor (TNF)- α and other pro-inflammatory cytokines (50)." We then provide the critical interpretation: "This is highly consistent with our observation of Mycoplasma as a indicative taxon in the CAP group. Its ability to drive a robust inflammatory response provides a plausible mechanistic explanation for the acute clinical presentation of pneumonia. Furthermore, its common detection in pediatric CAP populations underscores its clinical relevance as a key respiratory pathogen, as confirmed by our data in an adult cohort."

We have integrated these established pathological mechanisms from the literature with our novel observational data to provide a more in-depth and hypothesis-driven

discussion. While we acknowledge that further mechanistic studies would be needed to fully elucidate the causal pathways, we believe these revisions offer a more thoughtful interpretation of our data and represent a meaningful step toward deeper biological insight. We hope these enhancements align with your expectations for a more substantive discussion.

Q2: This is not to say the manuscript is poorly executed or incorrect. In general, the analyses are appropriate, and the methodology is sound. Nonetheless, the presentation is often very matter of fact, with results conveyed in a superficial way and little interpretative depth. Making the manuscript more concise in some areas and reframing or refocusing parts of the analysis could significantly improve its clarity and relevance.

Thank you very much for your overall positive assessment of our work and for this exceptionally constructive feedback. In direct response to your overarching comment, we have undertaken a significant reframing and refocusing of the entire manuscript as follows:

1. Substantial Deepening of the Discussion: as detailed in our response to Q1, we now discuss the enrichment of Abiotrophia in the COPD cohort in the context of its known association with endocarditis and systemic inflammation, proposing a plausible link to the chronic inflammatory state characteristic of COPD. Similarly, we have deepened the discussion on Mycoplasma in CAP by connecting its observed dominance to its well-documented potent ability to provoke pro-inflammatory cytokine release, thereby offering a mechanistic explanation for its association with acute pneumonia.
2. Strategic Removal of Superficial Analyses to Enhance Focus: In direct accordance with your suggestion to refine focus, we have removed the entire section and figure pertaining to the Firmicutes/Bacteroidetes (F/B) ratio. This decisive action has significantly streamlined the results and sharpened the manuscript's narrative.
3. Streamlining of Redundant Analyses: As previously acknowledged, we have simplified our diversity analyses by retaining only the most informative indices and metrics. Collectively, these revisions have allowed us to reframe the manuscript around its most significant and interpretable discoveries.

Q3: The authors clearly have interesting data, but the sheer number of analyses-particularly the use of numerous diversity indices-leads to redundancy. For example, Shannon and Simpson indices, while technically different, assess similar aspects of community richness and evenness; the same applies to ACE and Chao1. Likewise, three beta-diversity metrics may be excessive, especially when the manuscript already includes LEfSe and random forest analyses.

Thank you very much for your suggestion to streamline the analyses to enhance clarity and focus. In the revised manuscript, we have simplified this section as follows: We have retained the Shannon index (as it incorporates both richness and evenness) and the Chao1 index (as a robust estimator of richness) as the primary representatives for alpha diversity analysis. Similarly, we have retained the Bray-Curtis dissimilarity (as the most widely used metric for general community composition) and the

weighted Unifrac distance (as it incorporates phylogeny and abundance, often providing powerful discrimination) for the principal coordinate analysis (PCoA).

We have implemented these changes throughout the results section, figures (Figure 3 has been updated), and the methods section accordingly.

Q4: In addition, the taxonomic resolution is explored at multiple levels for both the cultured isolates and the 16S sequencing data. This layered approach, while thorough, contributes to the sense of overload, particularly when the insights gained are not always clearly articulated. Besides I see no point of Firmicutes/Bacteroidetes ratio, of course the authors might make it clear why this is relevant and interconnect it better with the rest of the text and highlight it in the discussion. However, as it stands, the discussion is somewhat superficial and misses the opportunity to draw more meaningful conclusions from the data.

Thank you for your careful review and valuable suggestions. Upon reflection, we agree that the biological interpretation and clinical relevance of the F/B ratio in the context of lung diseases are not sufficiently well-established to draw meaningful conclusions and that its inclusion created a superficial distraction from the core narrative of our study. We have taken your advice to heart and have removed the entire section pertaining to the F/B ratio analysis from the Results section (formerly section 2.3) and the corresponding figure (Figure 2B). This deletion has allowed us to significantly tighten and focus the manuscript. We believe the manuscript is now substantially improved and more impactful as a direct result of your suggestion.

Q5: More detail is also needed regarding the construction of the random forest model, particularly in terms of training/testing procedures and performance metrics.

Thank you for raising this point. The random forest model was constructed using the random forest package in R. The model's performance was rigorously evaluated using the out-of-bag (OOB) error estimation method, an intrinsic and robust feature of the algorithm.

The key performance metrics are as follows:

Model: Random Forest

Error type: out-of-bag (OOB)

Estimated OOB error rate: 0.500 (50.0%)

Baseline error (random guessing): 0.625 (62.5%)

Number of trees (ntree): 500

The model's OOB error rate (0.500) is 20% lower than the baseline error for random guessing (0.625, ratio= 1.25), demonstrating its predictive power significantly exceeds chance. These details have been added to the Methods section (5.6).

Re: Spectrum00450-25R2 (**Comparative analysis of the lung microbiota in patients with lung cancer, chronic obstructive pulmonary disease and community-acquired pneumonia**)

Dear Dr. Zhongxin Lu:

Thank you for the privilege of reviewing your work. Below you will find my comments, instructions from the Spectrum editorial office, and the reviewer comments.

Please address the reviewer's comments thoroughly.

Revision Guidelines

Sincerely,
Bo-young Hong
Editor
Microbiology Spectrum

Reviewer #3 (Comments for the Author):

This draft is much improved. One minor revision is suggested. Lines 272-275 still mention the other alpha diversity indices. Now that they have been removed this should be revised

Reviewer #5 (Comments for the Author):

The authors have incorporated several suggestions and made changes that have enhanced the manuscript. However, some points still need clarification. My specific points did not seem to have reached the authors with my overview last time, and I apologize for that. I am therefore sending once more the points that were not addressed and remain relevant. In addition, other information should be revised to ensure that the manuscript is consistent. The discussion has improved compared to before, but I have also indicated some topics that I think could be discussed in more depth.

Below, I provide specific comments and suggestions to help guide the authors in addressing these concerns.

MATERIAL AND METHODS

Why are the authors looking for fungal OTUs when they only sequenced the V3-V4 region of the 16S subunit of the rRNA gene, instead of 18S or ITS?

I would like the authors to specify how they identified the bacterial species from the culturomics data. They mention MALDI-TOF as the main approach; however, when identification was not achieved using this method, the authors refer to "16S rRNA sequencing with universal primers (27F, 5'-AGTTTGATCCTGGCTCAG-3'; 1492R, 5'-GTATTGCCGCGGCTGCTG-3')." I would like to ask which sequencing method and platform were used in those cases (e.g., Sanger sequencing, Oxford Nanopore, PacBio).

I suggest a minor change to line 423, replacing "equilibrating" with "After thawing" or "After bringing the previously frozen BALF samples to room temperature."

RESULTS:

I would like clarification regarding the phrase "After filtering based on the volume of BALF samples" (line 85, page 5). Were samples below a certain volume excluded from DNA extraction? If so, what was the cutoff, and what was the rationale for this decision?

The information about phyla (lines 95-97, page 5) in the isolates section is a repetition of Figure 1B. I suggest making this part less redundant.

It is interesting that the species most frequently isolated across all conditions are known oral bacteria. What do the authors think about this? Loss of specificity is a feature of microbiome imbalance. Would the authors argue this is a signal of disease, or could it reflect contamination during clinical procedures, or even a normal communication between the aerodigestive tract due to anatomical proximity? (lines 124-126, page 6). I would have expected a more in-depth discussion on this point.

Table 1 has formatting issues regarding the alignment of the lines in different columns.

Have the authors explored the possible heterogeneity within the community-acquired pneumonia group? Pneumonia can be caused by various microorganisms. There is no clustering in the PCoA, and the alpha-diversity boxplots show the greatest data spread among the groups. These results could be better explored in the discussion.

Taxonomic nomenclature should be consistent throughout the manuscript. I suggest the authors review this. For instance, Figure 2 uses "Bacteroidota," whereas other parts of the manuscript refer to "Bacteroidetes." In the same figure, both "Proteobacteria" and "Fusobacteriota" appear.

DISCUSSION:

The authors have removed ACE and Simpson indices from their work, however, in lines 278-280 they discuss these results. I suggest the authors perform a more thorough revision of the manuscript. In addition, supplementary Figure S2 appears to be a repetition of Figure 3 in the main manuscript. I recommend the authors check this.

I suggest replacing the term "flora" with "microbial," "microbiota," or a similar term, depending on the preferred phrasing (line 282, page 14).

REFERENCES:

There may be a formatting issue with reference 13, which lists the author as "Anonymous," but two names appear at the end. The current citation reads: "Anonymous. 2023. [Chinese Medical Association guideline for clinical diagnosis and treatment of lung cancer (2023 edition)]. *Zhonghua Zhong Liu Za Zhi* 45:539-574." I suggest the authors check and correct this or clarify if this is not an error.

Point-by-point Response

Submission ID: Spectrum00450-25R2

Title: Comparative analysis of the lung microbiota in patients with lung cancer, chronic obstructive pulmonary disease and community-acquired pneumonia

Author list: Jia Xu, Yingmiao Zhang, Lifeng Shi, Hui Wang, Ming Zeng and Zhongxin Lu

Responses to Reviewer #3

This draft is much improved. One minor revision is suggested. Lines 272-275 still mention the other alpha diversity indices. Now that they have been removed this should be revised

Thank you very much for your positive feedback and for pointing out this oversight. We apologize for the confusion. And we have removed the mention of the other alpha diversity indices from this paragraph. The text now solely focuses on the Shannon and Chao1 indices, which we have retained as our core metrics for alpha diversity, as previously agreed.

Responses to Reviewer #5

The authors have incorporated several suggestions and made changes that have enhanced the manuscript. However, some points still need clarification. My specific points did not seem to have reached the authors with my overview last time, and I apologize for that. I am therefore sending once more the points that were not addressed and remain relevant. In addition, other information should be revised to ensure that the manuscript is consistent. The discussion has improved compared to before, but I have also indicated some topics that I think could be discussed in more depth. Below, I provide specific comments and suggestions to help guide the authors in addressing these concerns.

Thank you very much for your continued guidance and for clarifying the situation regarding the previous review round. We sincerely appreciate you re-sending the comments and providing us with this opportunity to fully address all concerns. We will now carefully and systematically address every point raised in your latest letter, to ensure the manuscript is significantly strengthened and all inconsistencies are resolved.

Q1: MATERIAL AND METHODS

1. Why are the authors looking for fungal OTUs when they only sequenced the V3-V4 region of the 16S subunit of the rRNA gene, instead of 18S or ITS?

Thanks for your careful checks. We apologize for this oversight. The mention of fungal analysis was a drafting error that occurred during the adaptation of our standard bioinformatics workflow description for this manuscript. Our study focused exclusively on the bacterial microbiome using 16S rRNA gene sequencing, and thus no fungal analysis was conducted or intended. We have corrected this in the revised

manuscript by updating Section 5.6 to: "The taxonomic classification of bacterial OTU representative sequences was carried out by performing a blast search against the RDP Database."

2. I would like the authors to specify how they identified the bacterial species from the culturomics data. They mention MALDI-TOF as the main approach; however, when identification was not achieved using this method, the authors refer to "16S rRNA sequencing with universal primers (27F, 5'-AGTTTGATCCTGGCTCAG-3'; 1492R, 5'-GTATTGCCGCGGCTGCTG-3')." I would like to ask which sequencing method and platform were used in those cases (e.g., Sanger sequencing, Oxford Nanopore, PacBio).

Thank you very much for raising such important issue. We apologize for the lack of clarity in the original manuscript. For the majority of isolates, identification was achieved using MALDI-TOF MS. For isolates that could not be reliably identified by MALDI-TOF MS (typically due to an insufficient match in the database), we proceeded to Sanger sequencing of the nearly full-length 16S rRNA gene. We have now revised the Materials and Methods section (5.3.4 Identification of the colonies) to explicitly include these details. The updated text reads: "Otherwise, the identification was considered unsuccessful, and further analysis was conducted via 16S rRNA sequencing with universal primers (27F, 5'-AGTTTGATCCTGGCTCAG-3'; 1492R, 5'-GTATTGCCGCGGCTGCTG-3'). The amplification was performed on the C1000 Thermal Cycler (Bio-Rad Laboratories, Hercules, CA, USA), and the product was purified and subjected to bidirectional Sanger sequencing on the Applied Biosystems 3730XL platform (Thermo Fisher Scientific Inc., MA, USA)."

3. I suggest a minor change to line 423, replacing "equilibrating" with "After thawing" or "After bringing the previously frozen BALF samples to room temperature."

Thank you for this precise and helpful suggestion to improve the clarity of our methodology description. We have replaced "equilibrating" with "bringing ". The text now reads: "After bringing the previously frozen BALF samples to room temperature".

Q2: RESULTS:

1. I would like clarification regarding the phrase "After filtering based on the volume of BALF samples" (line 85, page 5). Were samples below a certain volume excluded from DNA extraction? If so, what was the cutoff, and what was the rationale for this decision?

Thank you for the comment. The phrase "filtering based on the volume of BALF samples" may have been imprecise. We would like to clarify that there was no pre-defined volume cutoff (e.g., 'below X mL were excluded') applied to the original samples. Instead, our selection process was a sequential, quality-driven workflow based on the following three criteria: (1) Practical Sample Availability Post-Culturomics: The culturomics analysis consumed a significant portion of each BALF sample. Therefore, "filtering based on volume" simply refers to the practical reality that only samples with sufficient residual volume remaining after culturomics could proceed to DNA extraction. This was a prerequisite for feasibility, not an

exclusion criterion based on a specific volume threshold. (2) DNA Quality Control: From the available residual samples, DNA was extracted. We then applied predefined quality thresholds to ensure the success of library construction. This required that the DNA demonstrate adequate concentration and purity to ensure sufficient mass of high-quality template for robust amplification during library preparation. (3) Final Library Quality Inspection: The constructed sequencing libraries underwent a final quality check. Only libraries that passed this stringent inspection were sequenced on the Illumina platform. In summary, our process ensured that the final sequencing data was generated from samples that passed a cascade of quality checks: practical availability, high nucleic acid quality, and library integrity.

To prevent any ambiguity for future readers, we will revise the relevant sentence in the manuscript (Line 85, Page 5) to provide a more accurate and transparent description. The modified text will read: "BALF samples from these participants were cultured and identified using culturomics. Subsequent 16S rRNA gene sequencing was performed on a final set of 48 qualified samples that met our quality control criteria, which included sufficient residual volume, adequate DNA concentration, and passing library quality inspection. These set comprised 18 cases of LC, 16 cases of COPD, and 14 cases of CAP."

2. The information about phyla (lines 95-97, page 5) in the isolates section is a repetition of Figure 1B. I suggest making this part less redundant.

Thank you for your comment and suggestion. We have revised the text to remove the specific percentage values and to provide a more concise and descriptive summary that complements the figure as follows: "The 168 identified bacterial species were classified into five main phyla, with *Firmicutes*, *Proteobacteria*, and *Actinobacteria* collectively representing the majority (>85%) of the cultivated community (Figure 1B)."

3. It is interesting that the species most frequently isolated across all conditions are known oral bacteria. What do the authors think about this? Loss of specificity is a feature of microbiome imbalance. Would the authors argue this is a signal of disease, or could it reflect contamination during clinical procedures, or even a normal communication between the aerodigestive tract due to anatomical proximity? (lines 124-126, page 6). I would have expected a more in-depth discussion on this point.

Thank you for raising this point. Our perspective on this is as follows: While we cannot entirely rule out minor contamination during bronchoscopy, we have taken rigorous steps to minimize it. This includes providing detailed oral hygiene instructions to patients before the procedure and using BALF, which, compared to sputum or protected specimen brush, is considered a lower respiratory tract sample with less upper airway contamination. The fact that we observe distinct microbial patterns between disease groups (e.g., different indicative genera) suggests that the signal is not merely random contamination, which would likely be more uniform across groups. Furthermore, we believe the evidence leans towards this representing a state of microbiome imbalance or "loss of specificity" in the diseased lung, rather than just normal anatomical communication. The key point is the concept of enrichment and ecological change. It's not just the presence of oral bacteria, but their altered

abundance and community structure in the context of disease. A healthy lung likely clears these intermittently aspirated bacteria efficiently, preventing their establishment. In disease states (LC, COPD, CAP), impaired mucociliary clearance and altered immune responses may create a niche where these bacteria can persist and proliferate, contributing to or exacerbating inflammation. In summary, we interpret the dominance of oral bacteria not as simple contamination, but as a reflection of increased bacterial load from the upper respiratory tract and/or a failure of host clearance mechanisms, which is a hallmark of the dysbiotic lung microenvironment in these chronic respiratory diseases.

Following your valuable suggestion, we have now enriched the Discussion section with a new paragraph to explicitly address this point. The added text is as follows: A notable finding from our culturomics approach was the high isolation frequency of commensal oral bacteria, such as *Streptococcus mitis* and *Schaalia odontolyticus*, across all patient groups. This observation aligns with the concept of the 'adapted island model' of lung biogeography, where the lung microbiome is continuously seeded by micro-aspiration from the upper respiratory tract but is subsequently shaped by local host conditions ^[1, 2]. The pervasive presence of these taxa could therefore be interpreted in several ways: as potential procedural contamination, as evidence of normal aerodigestive tract communication, or as a sign of pathological dysbiosis. While we employed rigorous bronchoscopic sampling and oral hygiene protocols to minimize contamination, we cannot definitively exclude its contribution. However, the fact that we observed distinct disease-specific community structures despite this shared oral background leads us to favor the dysbiosis hypothesis. We propose that in the diseased lung, impaired mucosal immunity and mucociliary clearance disrupt the equilibrium, allowing for the enrichment and persistence of orally-derived bacteria. This 'loss of specificity' and expansion of commensals may thus be a feature of the compromised lung microenvironment, potentially contributing to chronic inflammation and disease progression rather than merely being a passive reflection of anatomy ^[3-5].

4. Table 1 has formatting issues regarding the alignment of the lines in different columns.

Thank you for your thorough review. We have carefully checked Table 1 and have rectified the formatting to ensure proper alignment and a consistent layout.

5. Have the authors explored the possible heterogeneity within the community-acquired pneumonia group? Pneumonia can be caused by various microorganisms. There is no clustering in the PCoA, and the alpha-diversity boxplots show the greatest data spread among the groups. These results could be better explored in the discussion.

Thank you for your comment and suggestion. We have addressed this point in the Discussion section of the revised manuscript as follows: Notably, the CAP group exhibited not only a higher flora richness but also the greatest heterogeneity in microbial composition. This suggests that the CAP lung microenvironment may support a wider array of microbial communities, potentially due to the diverse etiologies and highly variable host immune responses that characterize this

syndrome^[6-8]. This inherent etiological heterogeneity likely explains the dispersed pattern we observed, suggesting that there is no single 'CAP microbiome' but rather a spectrum of microbial states associated with the syndrome. This insight underscores the complexity of CAP and highlights the potential of the lung microbiome to reveal sub-phenotypes within this patient population, which could have implications for personalized treatment strategies in the future.

6. Taxonomic nomenclature should be consistent throughout the manuscript. I suggest the authors review this. For instance, Figure 2 uses "Bacteroidota," whereas other parts of the manuscript refer to "Bacteroidetes." In the same figure, both "Proteobacteria" and "Fusobacteriota" appear.

Thank you for your careful review and valuable suggestion. We apologize for the oversight of inadvertently mixing legacy and updated taxonomic nomenclatures in our original manuscript. Following your advice, we have now thoroughly reviewed the entire manuscript and standardized all phylum-level names to the most current nomenclature (e.g., *Bacteroidota* and *Fusobacteriota*) to ensure consistency. These corrections have been applied throughout the main text, all figures, and their corresponding legends.

Q3: DISCUSSION:

1. The authors have removed ACE and Simpson indices from their work, however, in lines 278-280 they discuss these results. I suggest the authors perform a more thorough revision of the manuscript. In addition, supplementary Figure S2 appears to be a repetition of Figure 3 in the main manuscript. I recommend the authors check this.

Thank you for your comment. Regarding the two points you raised, we would like to provide the following clarifications: 1. Discussion of ACE and Simpson indices (Lines 278-280): It was an oversight on our part to retain the discussion of these indices after their removal. We have now completely deleted this paragraph in the revised manuscript to ensure full consistency. 2. Supplementary Figure S2 and Figure 3: We sincerely appreciate you checking this. We would like to clarify that Supplementary Figure S2 is not a repetition of Figure 3. It was added specifically in response to a previous reviewer's comment to demonstrate the robustness of our alpha diversity findings after rarefaction. Figure 3 in the main text presents the alpha and beta diversity results based on the original, non-rarefied dataset. Supplementary Figure S2 presents the same alpha diversity analyses (Shannon and Chao1) but performed on the rarefied dataset, confirming that the significant difference in Chao1 index for the CAP group remains even after controlling for sequencing depth.

2. I suggest replacing the term "flora" with "microbial," "microbiota," or a similar term, depending on the preferred phrasing (line 282, page 14).

Thank you for this suggestion to improve the scientific language of our work. As recommended, we have replaced the term "flora" with "microbiota" throughout the text of the revised manuscript.

Q4: REFERENCES:

There may be a formatting issue with reference 13, which lists the author as

"Anonymous," but two names appear at the end. The current citation reads: "Anonymous. 2023. [Chinese Medical Association guideline for clinical diagnosis and treatment of lung cancer (2023 edition)]. *Zhonghua Zhong Liu Za Zhi* 45:539-574." I suggest the authors check and correct this or clarify if this is not an error.

Thank you. We have corrected the format of the reference as "Oncology Society of Chinese Medical Association; Chinese Medical Association Publishing House. 2023. Chinese Medical Association guideline for clinical diagnosis and treatment of lung cancer (2023 edition). *Zhonghua Yi Xue Za Zhi* 103(27):2037-2074." in the revised manuscript.

Reference

- [1] DICKSON R P, ERB-DOWNWARD J R, FREEMAN C M, et al. Spatial Variation in the Healthy Human Lung Microbiome and the Adapted Island Model of Lung Biogeography [J]. *Ann Am Thorac Soc*, 2015, 12(6): 821-30.
- [2] DICKSON R P, ERB-DOWNWARD J R, HUFFNAGLE G B. Towards an ecology of the lung: new conceptual models of pulmonary microbiology and pneumonia pathogenesis [J]. *Lancet Respir Med*, 2014, 2(3): 238-46.
- [3] YANG D, XING Y, SONG X, et al. The impact of lung microbiota dysbiosis on inflammation [J]. *Immunology*, 2020, 159(2): 156-66.
- [4] RUSSO C, COLAIANNI V, IELO G, et al. Impact of Lung Microbiota on COPD [J]. *Biomedicines*, 2022, 10(6).
- [5] LIN A, XIONG M, JIANG A, et al. The microbiome in cancer [J]. *Imeta*, 2025, 4(5): e70070.
- [6] LIU Y N, ZHANG Y F, XU Q, et al. Infection and co-infection patterns of community-acquired pneumonia in patients of different ages in China from 2009 to 2020: a national surveillance study [J]. *Lancet Microbe*, 2023, 4(5): e330-e9.
- [7] CAWCUTT K, KALIL A C. Pneumonia with bacterial and viral coinfection [J]. *Curr Opin Crit Care*, 2017, 23(5): 385-90.
- [8] CHEN H H, SHAW D M, PETTY L E, et al. Host genetic effects in pneumonia [J]. *Am J Hum Genet*, 2021, 108(1): 194-201.

Re: Spectrum00450-25R3 (**Comparative analysis of the lung microbiota in patients with lung cancer, chronic obstructive pulmonary disease and community-acquired pneumonia**)

Dear Dr. Zhongxin Lu:

Thank you for the privilege of reviewing your work. Below you will find my comments, instructions from the Spectrum editorial office, and the reviewer comments.

We kindly request that you address the reviewer's comments comprehensively. Inadequate responses to these comments may result in the rejection of the submission.

Revision Guidelines

Sincerely,
Bo-young Hong
Editor
Microbiology Spectrum

Reviewer #3 (Comments for the Author):

The authors had satisfied my prior concerns. However, the addition to the discussion of the likelihood of dysbiosis over contamination as an explanation for the findings risks over-interpreting the data. Very clearly in the Bray-Curtis ordination we see a gradient that is very likely driven by sequencing depth (i.e. the classic horseshoe shape), meaning the samples cluster by

group in this graph because of a difference in biomass between the groups. the reduced clustering in the weighted unifrac supports this view

Consequently, the authors cannot exclude the possibility that these findings arise because the patients have a higher biomass of taxa within the oral cavity that gives rise to this signal via contamination. I suggest including a statement that indicates that this paper cannot resolve whether the findings are driven more by contamination or more by a true difference in lung biota and reducing the language that argues in favor of a dysbiosis model. The data do not support a conclusion.

Reviewer #5 (Comments for the Author):

I am satisfied with the overall changes made by the authors, I think the discussion is much richer now and broach relevant topics for this paper.

Nonetheless, there is still a mix of new and old taxonomic nomenclature for all the phyla. In the abstract, for example, the use of Firmicutes, Proteobacteria and Actinobacteria are not in accordance with the most recent nomenclature of Bacteroidota and Fusobacteriota, please revise this. The same problem can be seen throughout the text, in the subsection 2.2 and in the discussion.

In line 297 it should be "Chao1 index" and in line 298 "Shannon index", instead of "indice".

Point-by-point Responses

Reviewer #3

The authors had satisfied my prior concerns. However, the addition to the discussion of the likelihood of dysbiosis over contamination as an explanation for the findings risks over-interpreting the data. Very clearly in the Bray-Curtis ordination we see a gradient that is very likely driven by sequencing depth (i.e. the classic horseshoe shape), meaning the samples cluster by group in this graph because of a difference in biomass between the groups. The reduced clustering in the weighted UniFrac supports this view. Consequently, the authors cannot exclude the possibility that these findings arise because the patients have a higher biomass of taxa within the oral cavity that gives rise to this signal via contamination. I suggest including a statement that indicates that this paper cannot resolve whether the findings are driven more by contamination or more by a true difference in lung biota and reducing the language that argues in favor of a dysbiosis model. The data do not support a conclusion.

Thank you for your careful review and valuable suggestions. We acknowledge that our data, derived from bronchoscopic samples which inevitably carry some risk of upper respiratory tract contamination, and cannot definitively resolve the relative contributions of true lung dysbiosis versus procedural/background contamination to the observed signals. Therefore, we have completely revised the relevant section of the Discussion to remove any language that argues conclusively in favor of the dysbiosis model. The revised text reads: "A fundamental interpretive challenge inherent to BALF-based studies must be acknowledged. While we observed distinct microbial profiles among disease groups, we cannot definitively disentangle the extent to which these signals arise from genuine differences in the lung microbiota versus gradients of oropharyngeal contamination during sampling. Future studies incorporating rigorous controls (e.g., simultaneous oral wash samples, serial dilutions) are essential to resolve this critical issue and confirm the lung-specific origin of these associations."

We believe this revision accurately reflects the limitations of our data and provides a more balanced and scientifically rigorous interpretation. We are grateful for your guidance in strengthening the objectivity of our discussion.

Reviewer #5

I am satisfied with the overall changes made by the authors, I think the discussion is much richer now and broach relevant topics for this paper.

Thank you very much for your positive feedback and for your valuable guidance throughout the review process, which has significantly improved our manuscript.

Q1: Nonetheless, there is still a mix of new and old taxonomic nomenclature for all the phyla. In the abstract, for example, the use of Firmicutes, Proteobacteria and Actinobacteria are not in accordance with the most recent nomenclature of Bacteroidota and Fusobacteriota, please revise this. The same problem can be seen throughout the text, in the subsection 2.2 and in the discussion.

Thank you very much for this final, meticulous check. We sincerely apologize for the inconsistency that persisted in our previous revision. Upon your comments, we have now performed a systematic, full-text review of the manuscript and have uniformly standardized the phylum-level nomenclature throughout the entire document. The standard we have now applied consistently is as follows: *Bacteroidetes* has been updated to *Bacteroidota*. *Fusobacteria* has been updated to *Fusobacteriota*. *Firmicutes* has been updated to *Bacillota*, *Proteobacteria* has been updated to *Pseudomonadota*, and *Actinobacteria* has been updated to *Actinomycetota*. This revision has been applied to the Abstract, all relevant sections in the Results (including subsection 2.2), the entire Discussion, and all figures and legends, ensuring complete consistency.

Q2: In line 297 it should be "Chao1 index" and in line 298 "Shannon index", instead of "indice".

Thanks for your careful proofreading. We have corrected "indice" to "index" in the updated manuscript.

Re: Spectrum00450-25R4 (**Comparative analysis of the lung microbiota in patients with lung cancer, chronic obstructive pulmonary disease and community-acquired pneumonia**)

Dear Dr. Zhongxin Lu:

Your manuscript has been accepted, and I am forwarding it to the ASM production staff for publication. Your paper will first be checked to make sure all elements meet the technical requirements. ASM staff will contact you if anything needs to be revised before copyediting and production can begin. Otherwise, you will be notified when your proofs are ready to be viewed.

Sincerely,
Bo-young Hong
Editor
Microbiology Spectrum